# Lnc-TALC promotes O$^6$-methylguanine-DNA methyltransferase expression via regulating the c-Met pathway by competitively binding with miR-20b-3p

Pengfei Wu[1,3], Jinquan Cai[1,4], Qun Chen[1,3], Bo Han[1], Xiangqi Meng[1], Yansheng Li[2], Ziwei Li[1], Ruijia Wang[1], Lin Lin[1], Chunbin Duan[1], Chunsheng Kang[2,4] & Chuanlu Jiang[1,4]

Long noncoding RNAs (lncRNAs) have emerged as new regulatory molecules implicated in diverse biological processes, including therapeutic resistance. However, the mechanisms underlying lncRNA-mediated temozolomide (TMZ) resistance in glioblastoma (GBM) remain largely unknown. To illustrate the role of lncRNA in TMZ resistance, we induce TMZ-resistant GBM cells, perform a lncRNA microarray of the parental and TMZ-resistant cells, and find an unreported lncRNA in GBM, lnc-TALC (temozolomide-associated lncRNA in glioblastoma recurrence), correlated with TMZ resistance via competitively binding miR-20b-3p to facilitate c-Met expression. A phosphorylated AKT/FOXO3 axis regulated lnc-TALC expression in TMZ-resistant GBM cells. Furthermore, lnc-TALC increased MGMT expression by mediating the acetylation of H3K9, H3K27 and H3K36 in MGMT promoter regions through the c-Met/Stat3/p300 axis. In clinical patients, lnc-TALC is required for TMZ resistance and GBM recurrence. Our results reveal that lnc-TALC in GBM could serve as a therapeutic target to overcome TMZ resistance, enhancing the clinical benefits of TMZ chemotherapy.

[1] Department of Neurosurgery, the Second Affiliated Hospital of Harbin Medical University, Neuroscience Institute, Heilongjiang Academy of Medical Sciences, 150086 Harbin, China. [2] Department of Neurosurgery, Tianjin Medical University General Hospital, Lab of Neuro-oncology, Tianjin Neurological Institute, Key Laboratory of Post-Neuroinjury Neuro-repair and Regeneration in Central Nervous System, Ministry of Education and Tianjin City, 300052 Tianjin, China. [3]These authors contributed equally: Pengfei Wu, Qun Chen. [4]These authors jointly supervised: Jinquan Cai, Chunsheng Kang, Chuanlu Jiang. Correspondence and requests for materials should be addressed to J.C. (email: caijinquan@hrbmu.edu.cn) or to C.K. (email: kang97061@tmu.edu.cn) or to C.J. (email: jcl6688@163.com)

LncRNAs have emerged as an abundant and functionally diverse species of ncRNAs[1,2], which can mediate base-pairing interactions that guide lncRNA-containing complexes to specific RNA or DNA target sites[3–5]. LncRNAs have been described as pseudogenes that compete for miRNA binding, playing widespread roles in gene regulation and cellular processes[2,6]. For example, LINC00673 acts as a tumor suppressor, diminishing SRC-ERK oncogenic signaling. However, a G>A change at rs1111655237 in exon 4 of LINC00673 creates a target site for miR-1231 binding, decreases PTPN11 ubiquitination, attenuates the effect of LINC00673 in an allele-specific manner, conferring susceptibility to tumorigenesis[7], and indicating the importance of embedded miRNAs in lncRNAs regulating onco-genic signaling pathways. Emerging evidence has revealed that lncRNAs, as competitive RNAs[6,8], mediate postoperative treat-ment resistance in some cancers[9]. Lnc-RI, a radiation-inducible lncRNA molecule involved in radiation-induced DNA damage response, acted as a ceRNA to stabilize RAD51 mRNA via competitively binding with miR-193a-3p and releasing of its inhibition on RAD51 expression[9]. Thus, the transcriptome pro-filing alteration of lncRNAs still needs to be illustrated in resistant tumor cells.

Glioblastoma (GBM) is the most common malignant primary brain cancer in adults, with a median survival of 14.6 months upon diagnosis[10,11], and a 5-year survival rate of only 5.5%[12]. This poor prognosis is due to therapeutic resistance and tumor recurrence following surgical removal, and the treatment of such brain tumors remains a challenge[13]. The alkylating drug TMZ is routinely used in brain tumor patients[10,14], but the major hurdle in GBM treatment is the development of resistance to TMZ chemotherapy. The lncRNA MALAT1 can promote TMZ resis-tance in GBM, and targeting MALAT1 sensitizes GBM to TMZ. The lncRNA-regulated TMZ-resistant mechanisms in GBM represent a crucial nodal point for therapeutic intervention[15–17]. Thus, it is urgent to elucidate the underlying lncRNA-based mechanisms of TMZ resistance in GBM patients.

Receptor protein tyrosine kinases (RTKs) are essential enzymes in cellular signaling processes that can regulate cell growth, dif-ferentiation, migration, and metabolism[18]. Activation of c-Met enhances GBM cell migration and tumor cell resistance in response to DNA damage[19,20]. In cancer cells, aberrant c-Met axis activation, closely related to c-Met gene mutations, over-expression, and amplification, promotes tumor development and progression by stimulating the PI3K/AKT[21], Ras/MAPK[22], JAK/STAT[23], SRC[24], and Wnt/β-catenin[25] signaling pathways, among others[26,27]. Therefore, c-Met and its associated signaling path-ways are clinically important therapeutic targets[28]. Few studies have investigated how the c-Met signaling pathway interacts with lncRNAs to contribute to TMZ resistance in GBM.

The DNA repair enzyme $O^6$-methylguanine-DNA methyl-transferase (MGMT) expression is lost in TMZ-responsive glio-mas and is highly expressed in TMZ-resistant gliomas[29]. Alkylating chemotherapy is a mainstay in the treatment of GBM despite primary and acquired resistance[30]. MGMT efficiently removes alkylating lesions at the $O^6$ position of guanine and repairs the DNA damage induced by DNA alkylators or chlor-oethylating agents, thereby causing treatment failure[31]. Although higher MGMT expression levels are accompanied by the devel-opment of TMZ resistance in GBM cells[32], the mechanism of MGMT upregulation in TMZ-resistant GBM cells has not been clarified.

In the present study, we investigate the contribution of lncRNAs by profiling alterations in TMZ resistance and explore the therapeutic implications of the lncRNA lnc-TALC in TMZ-resistant GBM cells. Our results show that lnc-TALC regulates the c-Met signaling pathway via competitively binding to miR-20b-3p

and activating the Stat3/p300 complex to promote MGMT expression and TMZ resistance by modulating the acetylation of histone H3.

## Results

**Lnc-TALC is highly expressed in TMZ-resistant GBM cells.** Patient-derived GBM cells 551W and HG7 were isolated from discarded GBM specimens. Four types of GBM cells, including LN229, U251, 551W, and HG7, were exposed to increasing TMZ concentrations and underwent cycles of TMZ treatment for 5 months. The GBM cells acquired TMZ resistance and were named 229R, 251R, 551WR, and HG7R (Fig. 1a). Compared with the parental cells, the TMZ-resistant cells exhibited a poor response to TMZ, as illustrated by an increased the half maximal inhibitory concentration (IC50), enhanced independent growth ability and decreased apoptosis under TMZ treatment (Supplementary Fig. 1a–c). The lncRNA microarrays were performed to compare lncRNA expression profiles between LN229 and 229R cells, and the top 30 up- or downregulated lncRNAs of TMZ-resistant GBM cells out of the entire genome were identified; the associations with mRNAs are shown in the circos plot (Fig. 1b). The heatmap shows the expression levels of the top 30 up- or downregulated lncRNAs (Fig. 1c). Single-sample gene set enrichment analysis (ssGSEA) illustrated that the top 30 upregulated lncRNAs were associated with the Gene Ontology (GO) and KEGG pathways enriched in the regulation of DNA repair and the MAPK signaling pathway, among others (Fig. 1d). The top 10 upregulated lncRNAs between LN229 and 229R cells are listed in the volcano plot (Fig. 1e), and were subjected to validation by quantitative real time polymerase chain reaction (qRT-PCR) using four groups of parental and resistant GBM cells (Supplementary Fig. 2a). RNAi was then used to knock down the 10 selected lncRNAs for further loss-of-function analysis in the TMZ-resistant GBM cells (Supplementary Fig. 2b). Knock-down of lnc-TALC (ENST00000424980.5) and NR_028415 both inhibited the TMZ resistance in two types of resistant cells (Fig. 2a). Lnc-TALC is composed of two exons with a full length of 418 nt determined by RACE (rapid amplification of cDNA ends) assay (Supplementary Fig. 2c). The non-coding nature of lnc-TALC was confirmed by coding-potential analysis (Supplementary Fig. 2d, e). To further elucidate the functional role of lnc-TALC in TMZ resistance, we observed that lnc-TALC was correlated with the ERK1 and ERK2 cascade, DNA repair, and the MAPK signaling pathway through GO and KEGG pathway analysis (Fig. 2b). We stably knocked down lnc-TALC in TMZ-resistant cells or over-expressed lnc-TALC in parental cells using the CRISPR-Cas9 system or LV-lnc-TALC, respectively (Supplementary Fig. 2f, g). Knock-down of lnc-TALC in resistant GBM cells significantly decreased cell viability, promoted cell apoptosis, and inhibited cell colony formation and proliferation after TMZ treatment (Fig. 2c–f). Overexpression of lnc-TALC in LN229 and HG7 cells led to a marked increase in TMZ IC50, inhibited cell apoptosis, and pro-moted cell proliferation and colony formation after TMZ treatment (Supplementary Fig. 2h–k). In addition, knockdown of lnc-TALC in resistant GBM cells obviously inhibited the phosphorylation of Stat3, AKT, and MAPK in TMZ-resistant cells. The overexpression of lnc-TALC in LN229 and HG7 cells promoted the phosphoryla-tion of Stat3, AKT, and MAPK (Fig. 2g), suggesting that lnc-TALC might function as part of the tyrosine kinase signaling pathway, which is involved in TMZ resistance of GBM cells.

**Phosphorylated AKT/FOXO3 axis regulates lnc-TALC expression.** We next investigated the underlying mechanism of lnc-TALC upregulation in TMZ-resistant GBM cells. We did not observe copy number aberrance of the lnc-TALC gene in the TMZ-resistant cell genomes (Supplementary Fig. 3a). Inhibition

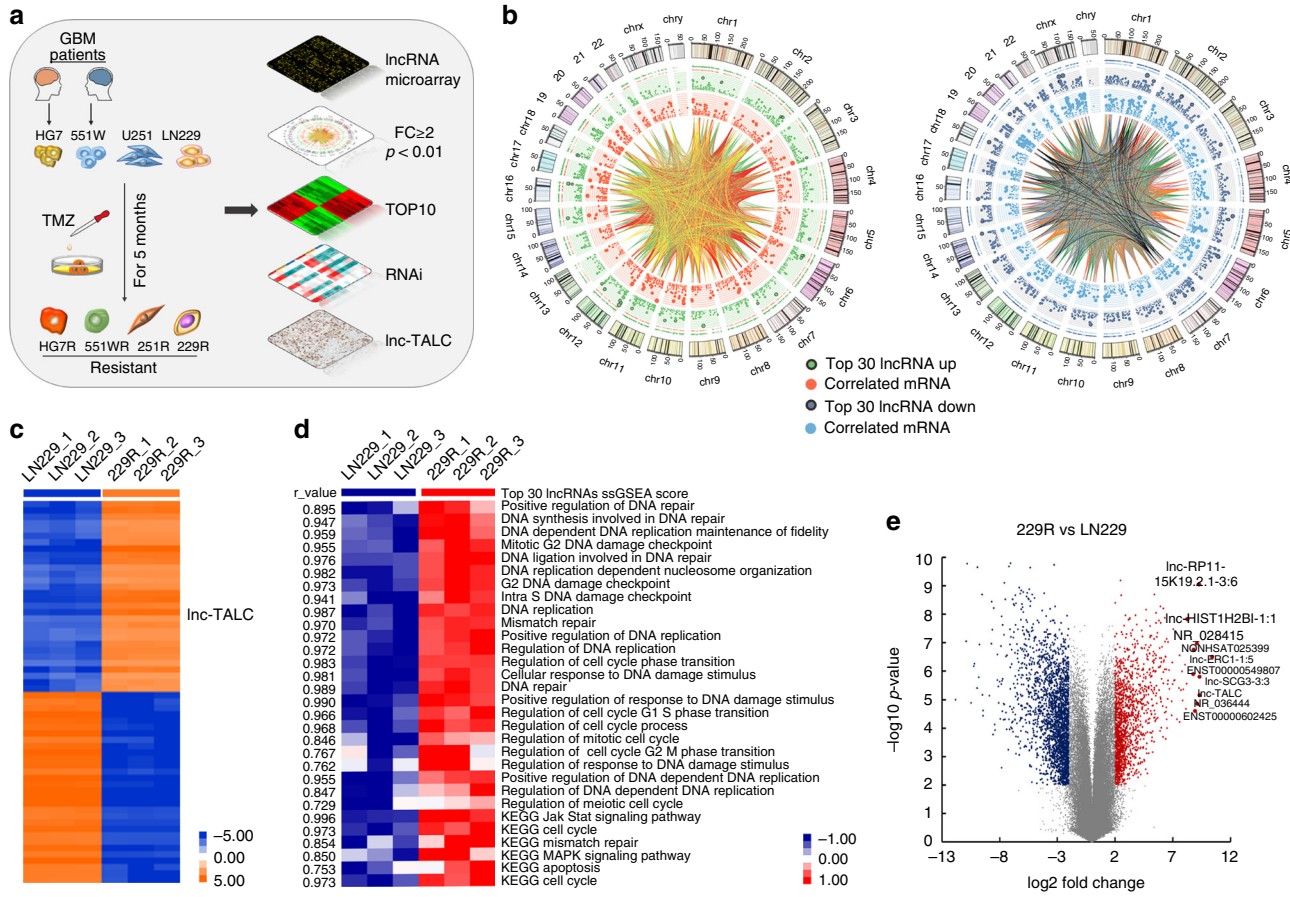

**Fig. 1** LncRNA microarray detection of TMZ-resistant and parental GBM cells. **a** Schematic process to acquire TMZ-resistant GBM cells and to screen the lncRNA lnc-TALC. **b** The circos plot presents the genome-wide view of the top 30 up- or downregulated lncRNAs and the associations with mRNAs of TMZ-resistant GBM cells. **c** The heatmap shows the top 30 up- and downregulated lncRNAs in TMZ-resistant GBM cells. **d** The top 30 upregulated lncRNAs associated with functional gene sets of TMZ-resistant GBM cells are shown in the heatmap. **e** The top 10 upregulated lncRNAs between LN229 and 229R cells are listed in the volcano plot

of DNA methyltransferase also had no effect on the lnc-TALC expression in GBM cells (Supplementary Fig. 3b).

Since activated AKT induced phosphorylation and cytoplasmic retention of the FOXO family for subsequent proteasome degradation[33], we performed a bioinformatics analysis of the promoter region of the lnc-TALC gene and predicted two DNA-binding elements (DBEs) for FOXO3 based on the FOXO families' binding sites in the JASPAR database[34,35] (Fig. 3a and Supplementary Table 1). The immunofluorescence assay showed that there was less aggregated FOXO3 in the nucleus of TMZ-resistant GBM cells than in parental cells (Fig. 3b). Transfection of constitutively active FOXO3 significantly downregulated lnc-TALC levels in TMZ-resistant cells, whereas DNA-binding domain-truncated mutants (FOXO3-Mut) had no effect on lnc-TALC levels (Fig. 3c). Knockdown of AKT in TMZ-resistant cells decreased lnc-TALC but increased the level and nuclear translocation of FOXO3 (Fig. 3d, e). We performed chromatin immunoprecipitation (ChIP)-PCR assay to detect the enrichment of FOXO3 on the promoter region of lnc-TALC, whereas the enrichment was significantly decreased in TMZ-resistant cells (Fig. 3f). The histone deacetylase inhibitors, suberoylanilide hydroxamic acid (SAHA) and sodium butyrate (NaB), abolished the FOXO3-triggered transcriptional suppression of lnc-TALC (Fig. 3g).

**Met regulated by lnc-TALC is required for TMZ resistance.** To further illuminate the potential mechanism of TMZ resistance, we

then selected 56 patients diagnosed with GBM and treated by TMZ chemotherapy from the Cancer Genome Atlas (TCGA) RPPA protein expression database. We found higher levels of AKT-pS473-R-V ($p = 0.0236$) and c-Met_pY1235-R-C ($p = 0.0485$) in samples with shorter survival times (Supplementary Fig. 4a). Moreover, we observed that the expression levels of RTK molecules were much higher in 229R cells through analyzing mRNA microarray profiles (Fig. 4a). We also investigated 502 GBM samples, including 483 primary GBM samples and 19 recurrent GBM samples, from the TCGA database. Differentially expressed genes were selected (FDR < 0.05), and MET had a higher level in recurrent GBM samples (Supplementary Fig. 4b). TMZ-resistant GBM cells showed a higher level of c-Met and p-Met than did parental GBM cells (Fig. 4b). We observed the effects of lnc-TALC on Met expression in TMZ-resistant GBM cells by knocking down lnc-TALC and parental GBM cells by overexpressing lnc-TALC. Knockdown of lnc-TALC decreased the c-Met and p-Met levels, and overexpression of lnc-TALC increased the c-Met and p-Met levels (Fig. 4c–f). Overexpression of MET restored the resistant phenotype and downstream signaling in TMZ-resistant GBM cells with knocked-down lnc-TALC (Fig. 4g, h and Supplementary Fig. 4c, d). Conversely, knockdown of MET inhibited the resistant phenotype and the downstream signaling in parental GBM cells overexpressing lnc-TALC (Fig. 4i, j and Supplementary Fig. 4e, f). Consistently, SGX-523, a small-molecule inhibitor of c-Met, restored the sensitivity and the downstream signaling of parental GBM cells after the

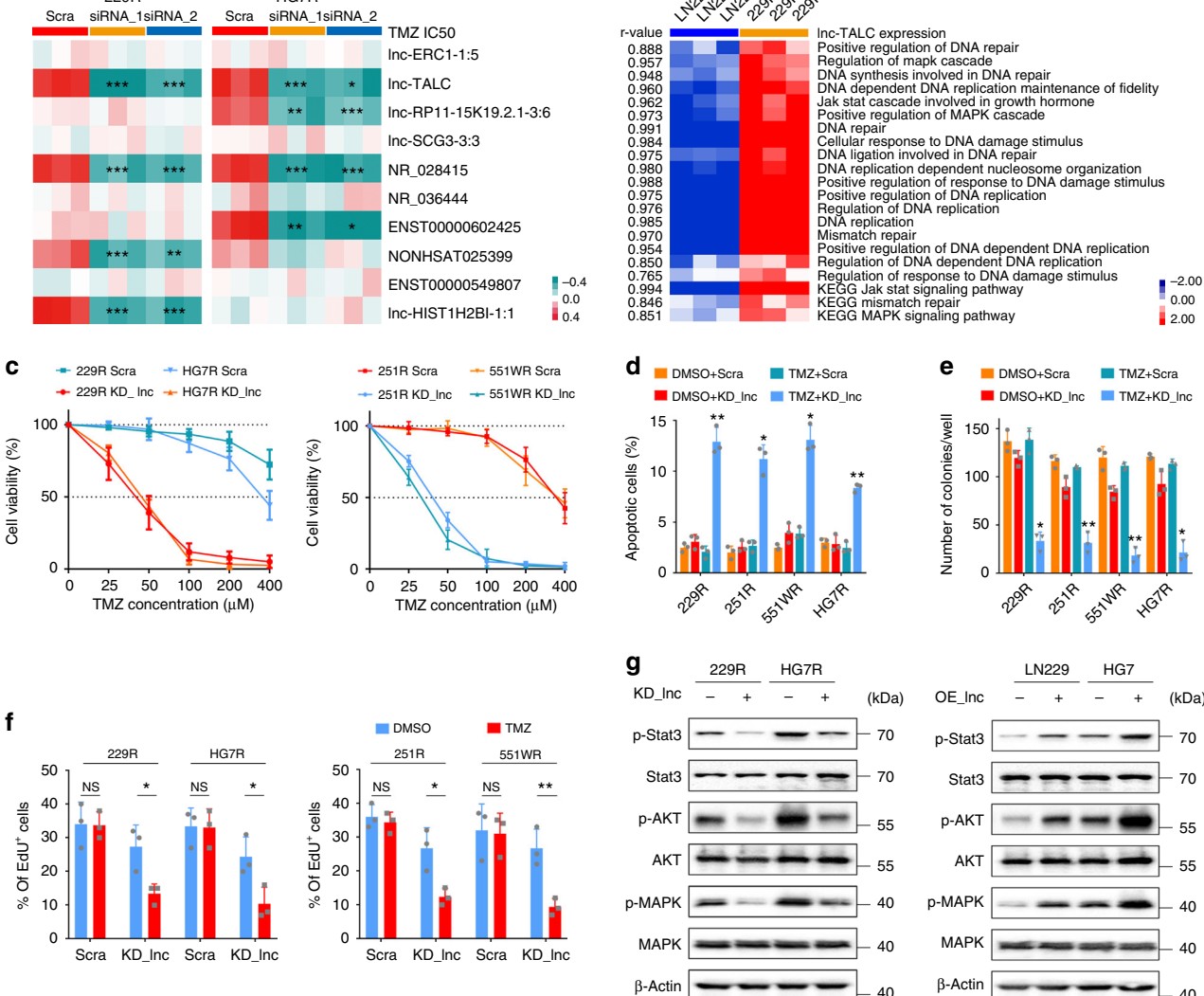

**Fig. 2** Knockdown of lnc-TALC restores TMZ sensitivity in TMZ-resistant GBM cells in vitro. **a** Determination of TMZ IC50 by CCK-8 assay analysis after knockdown of the top 10 upregulated lncRNAs in 229R and HG7R cells upon TMZ treatment (100 μM) at the indicated concentrations for 72 h (n = 3). **b** Association heatmap between lnc-TALC expression and functional gene sets. **c** CCK-8 assay analysis revealed the effect of lnc-TALC knockdown on the TMZ-resistant cells with TMZ treatment at the indicated concentrations for 72 h (n = 3). **d** Flow cytometric analysis revealing the effect of lnc-TALC knockdown on the apoptosis of TMZ-resistant cells with TMZ treatment (100 μM) for 72 h (n = 3). **e** Colony formation assay detecting the effect of lnc-TALC knockdown on the growth of TMZ-resistant cells with TMZ treatment in a 6-well dish (300 cells per well) for 11 days (n = 3). Representative images and the average number of colonies are shown. **f** EdU assay showing the effect of lnc-TALC knockdown on DNA replication activity of TMZ-resistant cells with TMZ treatment (100 μM) for 72 h (n = 3). **g** Western blot analysis of indicated proteins in TMZ-resistant and parental GBM cells transfected with LV-CRISPR-lnc-TALC, LV-lnc-TALC or scramble. Data are presented as the mean ± S.D. P value was determined by Student's t-test or one-way ANOVA. Significant results were presented as NS non-significant, *P < 0.05, **P < 0.01, or ***P < 0.001

overexpression of *lnc-TALC* (Fig. 4k, l and Supplementary Fig. 4g). Taken together, these results indicate that inhibition of the c-Met signaling pathway reversed the *lnc-TALC*-induced TMZ resistance in GBM cells.

**Lnc-TALC competitively binds miR-20b-3p targeting MET 3′ UTR.** In addition to epigenetic regulation in the nucleus, lncRNAs can also regulate target gene expression by functioning as competing endogenous RNAs (ceRNAs) for specific miRNAs in the cytoplasm. *Lnc-TALC* was identified as a cytoplasm-enriched abundant lncRNA (Supplementary Fig. 5a). Suppression of dicer increased the expression of c-Met (Supplementary Fig. 5b). The RNA immunoprecipitation (RIP) schematic process assay showed that *lnc-TALC* and c-MET transcript could bind Ago2 (Fig. 5a). We overexpressed *lnc-TALC* in LN229 cells and

observed increasing enrichment of *lnc-TALC* and decreasing enrichment of c-MET transcript in Ago2 compared to controls. We also knocked down *lnc-TALC* in 229R cells and observed decreasing enrichment of *lnc-TALC* and increasing enrichment of c-MET transcript in Ago2 compared with controls. (Fig. 5b).

Based on the DIANA, RegRNA2.0, TargetScan, miRWalk, and miRNA.org databases (Supplementary Table 2), two candidate miRNAs (*miR-20b-3p* and *miR-335-3p*) were predicted to target both the *lnc-TALC* and *MET* 3′UTR regions (Supplementary Fig. 5c). Furthermore, the dual luciferase reporter assays (Fig. 5c and Supplementary Fig. 5d) revealed that only *miR-20b-3p* directly bound to the *lnc-TALC* and *MET* 3′UTR regions (Fig. 5d and Supplementary Fig. 5e). The MS2-RIP assay was conducted to pull down endogenous miRNAs associated with *lnc-TALC* and showed that *miR-20b-3p* was significantly enriched in RNAs

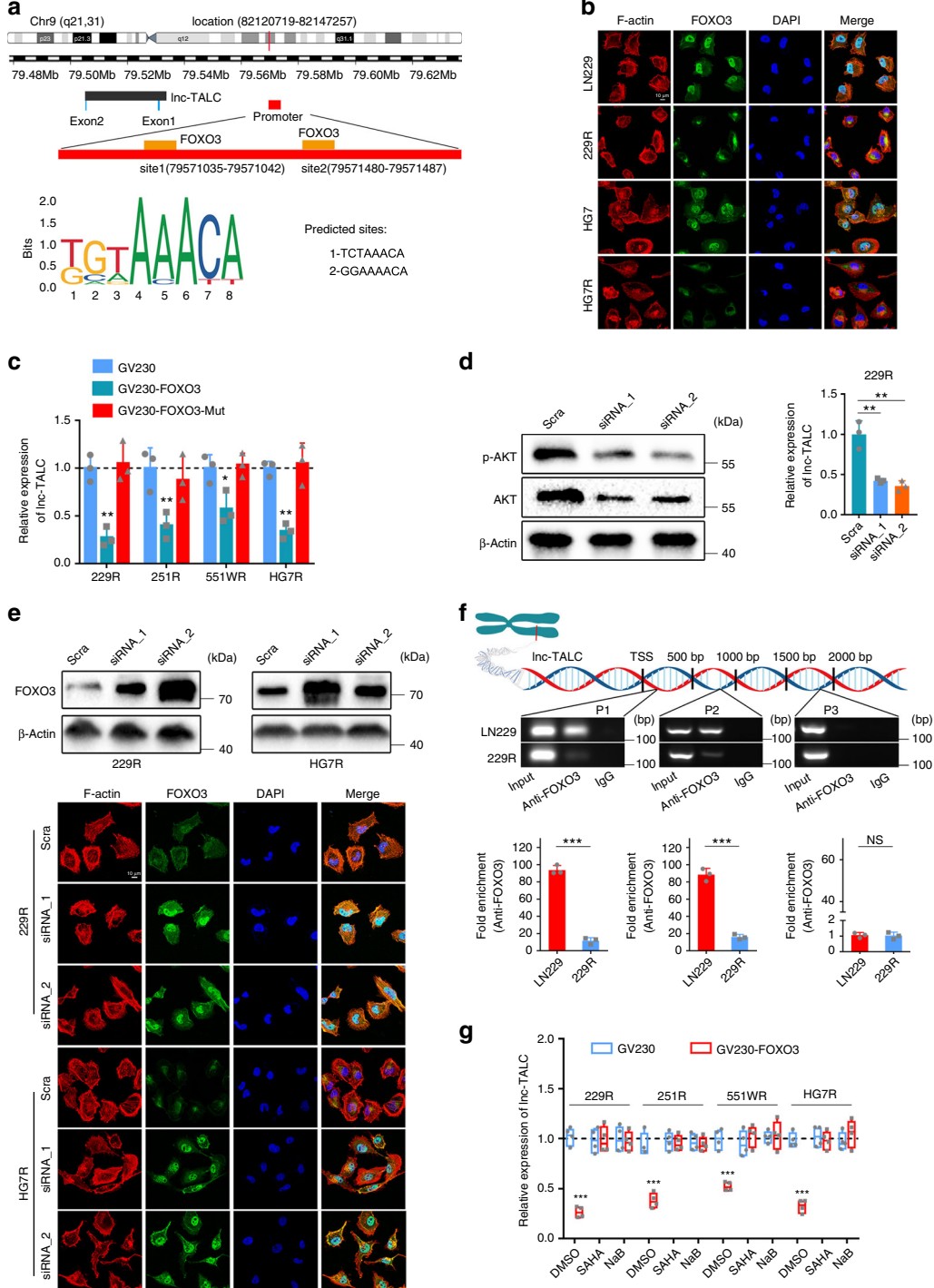

**Fig. 3** Phosphorylated AKT and the FOXO3 axis regulates lnc-TALC expression in TMZ-resistant GBM cells. **a** Predicted FOXO3-binding sites in the promoter region of lnc-TALC. **b** Immunofluorescence analysis of FOXO3 in TMZ-resistant and parental GBM cells. The nuclei were stained with DAPI. Scale bar = 10 μm. **c** qRT-PCR analysis of lnc-TALC levels in TMZ-resistant GBM cells transfected with FOXO3 and FOXO3-Mut plasmids after 72 h. **d** Left: Western blot analysis of indicated proteins in 229R cells transfected with scramble or AKT siRNA after 72 h. Right: qRT-PCR analysis of lnc-TALC in AKT siRNA and scramble TMZ-resistant GBM cells. **e** Upper: Western blot analysis of FOXO3 in TMZ-resistant GBM cells transfected with scramble or AKT siRNA after 72 h. Lower: Immunofluorescence analysis of FOXO3 in AKT siRNA and scramble TMZ-resistant GBM cells. The nuclei were stained with DAPI. Scale bar = 10 μm. **f** ChIP-PCR assay of the enrichment of FOXO3 on the lnc-TALC promoter region normalized to IgG in LN229 and 229R cells ($n$ = 3). P3 served as a negative control without DBEs for FOXO3. **g** qRT-PCR analysis of lnc-TALC in TMZ-resistant GBM cells treated with SAHA and NaB after transfection with GV230-FOXO3 ($n$ = 4). Data are presented as the mean ± S.D. $P$ value was determined by Student's $t$-test or one-way ANOVA. Significant results were presented as NS non-significant, *$P$ < 0.05, **$P$ < 0.01, or ***$P$ < 0.001

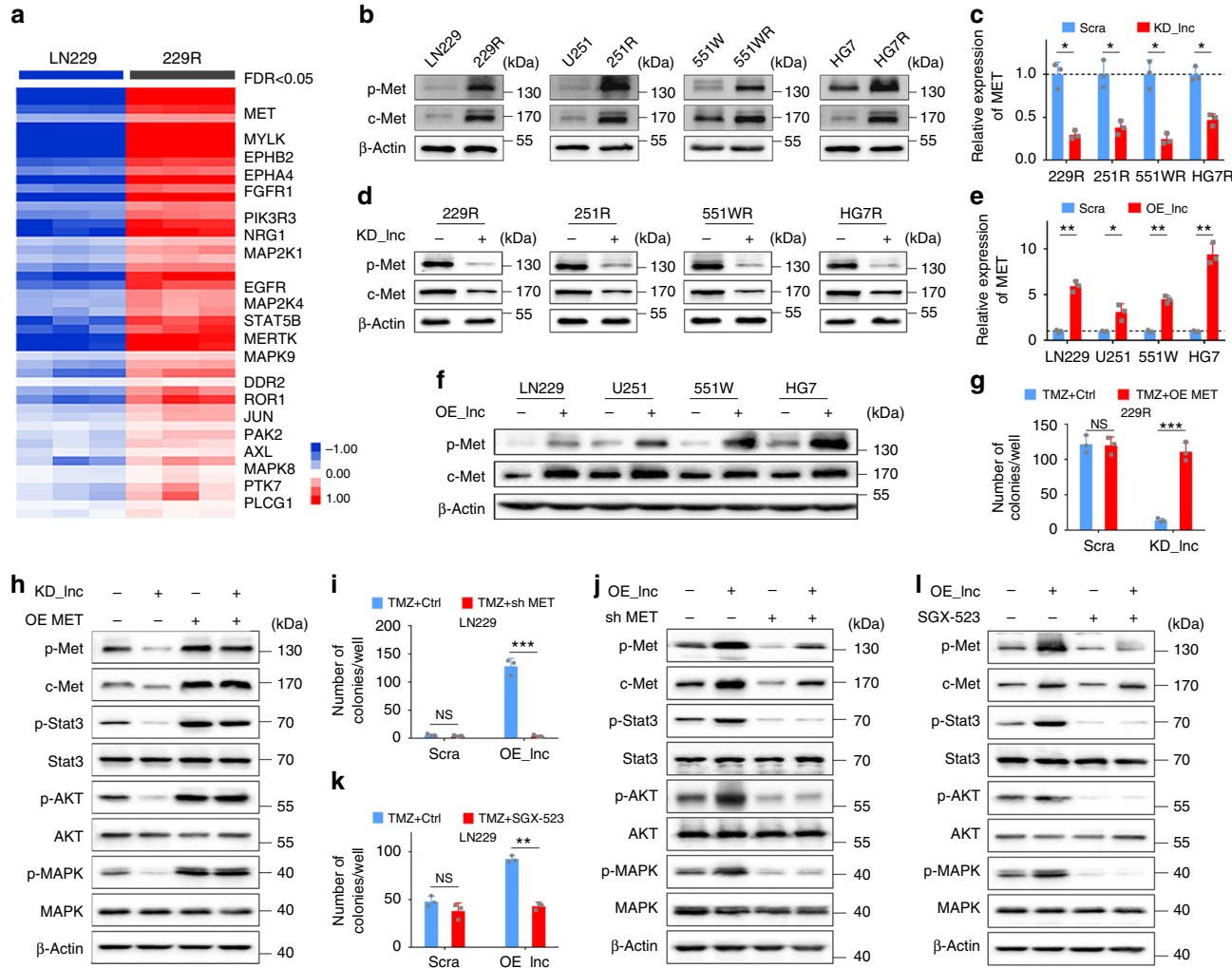

**Fig. 4** Met regulated by lnc-TALC is highly expressed in recurrent GBM and is required for TMZ resistance. **a** The levels of RTK molecules were analyzed in mRNA microarray profiles. **b** Western blot analysis of p-Met and c-Met levels in TMZ-resistant and parental GBM cells. **c** qRT-PCR analysis of the effect of lnc-TALC knockdown on MET expression in TMZ-resistant GBM cells. **d** Western blot analysis of c-Met and p-Met levels affected by lnc-TALC knockdown in TMZ-resistant GBM cells. **e** qRT-PCR analysis of the effect of lnc-TALC overexpression on MET expression in parental GBM cells. **f** Western blot analysis of c-Met and p-Met levels affected by lnc-TALC overexpression in parental GBM cells. **g** Colony formation assay of the effect of MET overexpression on 229R knock down of lnc-TALC with TMZ treatment (100 μM) in a six-well dish (300 cells per well) for 11 days (n = 3). The average number of colonies is shown. **h** Western blot analysis of the indicated proteins in 229R GBM cells after lnc-TALC knockdown or MET overexpression. **i** Colony formation assay of the effect of sh-MET on LN229 overexpressing lnc-TALC with TMZ treatment (100 μM) in a 6-well dish (300 cells per well) for 11 days (n = 3). The average number of colonies is shown. **j** Western blot analysis of the indicated proteins in LN229 GBM cells after lnc-TALC overexpression or MET knockdown. **k** Colony formation assay of the effect of SGX-523 (200 nM) on LN229 cells overexpressing lnc-TALC with TMZ treatment (100 μM) in a six-well dish (300 cells per well) for 11 days (n = 3). The average number of colonies is shown. **l** Western blot analysis of the indicated proteins in LN229 GBM cells after SGX-523 (200 nM) treatment. Data are presented as the mean ± S.D. P value was determined by Student's t-test. Significant results were presented as NS non-significant, *P < 0.05, **P < 0.01, or ***P < 0.001

retrieved from MS2bs-*lnc-TALC*, further confirming that *miR-20b-3p* specifically targeted *lnc-TALC* (Fig. 5e and Supplementary Fig. 5f). Then, qRT-PCR was used to quantify the molecular numbers of *lnc-TALC, MET*, and *miR-20b-3p* per cell (Supplementary Fig. 5g). The dual luciferase reporter assays showed that *miR-20b-3p* bound the *MET* mRNA 3′UTR sequence in parental or TMZ-resistant GBM cells (Fig. 5f and Supplementary Fig. 5h, i). Moreover, c-Met and its downstream activity were restored after *miR-20b-3p* was inhibited in *lnc-TALC* knockdown cells. In contrast, the overexpression of *miR-20b-3p* inhibited c-Met and its downstream expression (Fig. 5g, h and Supplementary Fig. 5j–l). The CCK-8 and colony formation assays showed that *miR-20b-3p* was involved in *lnc-TALC*-mediated TMZ resistance in GBM cells (Supplementary Fig. 5m–p).

**Lnc-TALC increases MGMT by modulating H3 acetylation.** MGMT, which is a DNA repair enzyme that quickly removes adducts from $O^6$-meG and repairs damaged guanine, mediates TMZ resistance in GBM[36]. In the present study, we confirmed that the mRNA and protein levels of MGMT were much higher in TMZ-resistant GBM cells than in parental GBM cells (Supplementary Fig. 6a and Fig. 6a). Repression of MGMT restored TMZ sensitivity in the TMZ-resistant GBM cells (Supplementary Fig. 6b). We then investigated whether *lnc-TALC* was responsible for the upregulation of MGMT. Overexpression of *lnc-TALC* in parental GBM cells increased the protein level of MGMT (Supplementary Fig. 6c). However, the MGMT level decreased when *lnc-TALC* was knocked down in TMZ-resistant GBM cells (Fig. 6b). Our previous results confirmed that *lnc-*

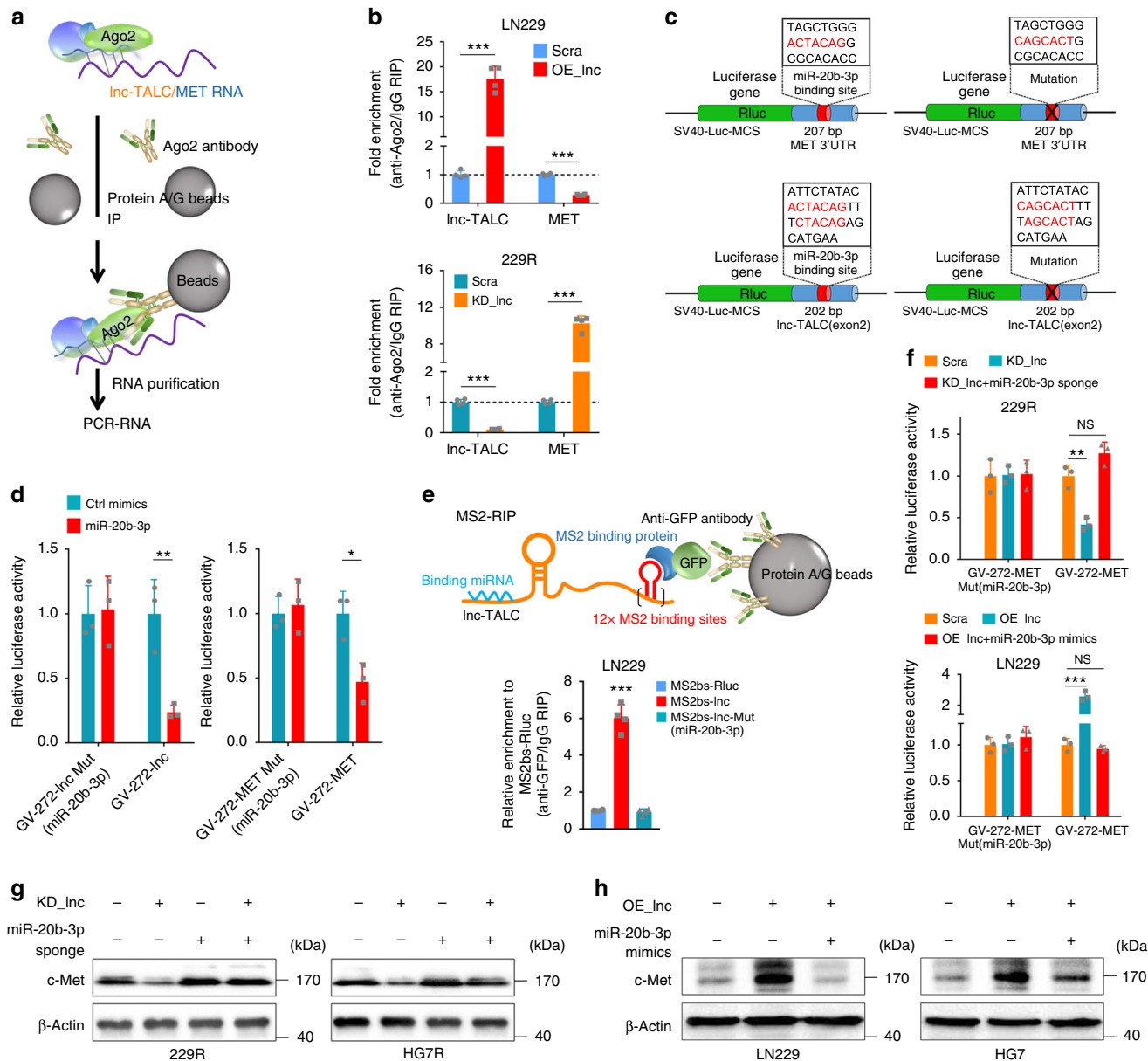

**Fig. 5** Lnc-TALC competitively binds miR-20b-3p targeting the MET 3′UTR region. **a** Schematic process of the RIP-PCR assay. **b** RIP-PCR assay of the enrichment of Ago2 on lnc-TALC and c-MET transcript normalized to IgG in LN229 or 229R cells transfected with LV-scramble, or LV-lnc-TALC or LV-CRISPR-lnc-TALC (n = 4). **c** Schematic outline of predicted and mutant binding sites for miR-20b-3p on lnc-TALC and MET. **d** Luciferase activity of GV-272-lnc-TALC or GV-272-MET upon transfection of control mimics and miR-20b-3p in 229R (n = 3). **e** Upper: Schematic process of the MS2-RIP assay. Lower: MS2-based RIP assay with anti-GFP antibody in LN229 cells 72 h after transfection with MS2bs-Rluc, MS2bs-lnc-TALC or MS2bs-lnc-TALC-Mut (n = 4). **f** Upper: Luciferase activity of GV-272-MET in 229R lnc-TALC knockdown upon transfection of a miR-20b-3p sponge (n = 3). Lower: Luciferase activity of GV-272-MET in LN229 overexpressing lnc-TALC upon transfection of miR-20b-3p mimics (n = 3). **g** Western blot analysis of c-Met in 229R and HG7R cells transfected with LV-CRISPR-lnc-TALC or a miR-20b-3p sponge. **h** Western blot analysis of c-Met in LN229 and HG7 cells transfected with LV-lnc-TALC or miR-20b-3p mimics. Data are presented as the mean ± S.D. P value was determined by Student's t-test or one-way ANOVA. Significant results were presented as NS non-significant, *P < 0.05, **P < 0.01, or ***P < 0.001

TALC promoted c-Met expression in TMZ-resistant GBM cells. The MGMT protein level was suppressed after c-Met was restrained (Fig. 6c and Supplementary Fig. 6d). The over-expression of c-Met promoted MGMT expression and phosphorylation of Stat3 and MAPK (Supplementary Fig. 6e, f). Inhibition or overexpression of *miR-20b-3p* promoted or suppressed the MGMT expression levels, respectively (Supplementary Fig. 6g). Furthermore, the corepression of Stat3 and MAPK markedly lowered TMZ resistance (Supplementary Fig. 6h).

It was necessary to further investigate how the c-Met signaling pathway regulates MGMT expression in TMZ-resistant GBM cells. The pyrosequencing assay showed no significant change in the methylation levels of the MGMT gene promoter region between parental cells and TMZ-resistant cells or between the *lnc-TALC* knockdown and control groups (Supplementary Fig. 7a). In public databases, multiple miRNAs that were reported to posttranscriptionally regulate MGMT expression did not exhibit a significant differential expression between recurrent GBMs and primary GBMs (Supplementary Fig. 7b). We also did not observe

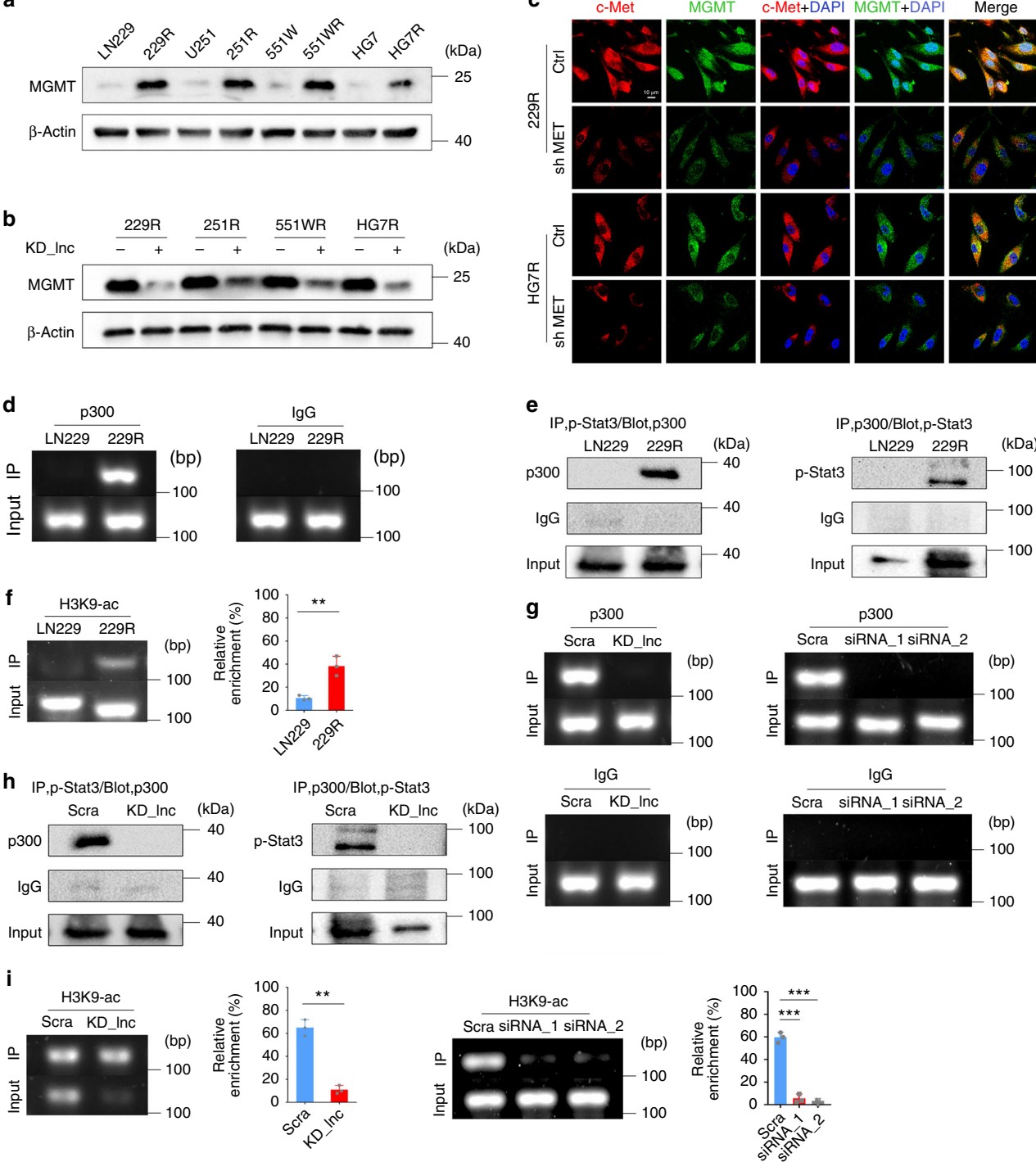

**Fig. 6** The Lnc-TALC/c-Met axis activating the Stat3/p300 complex increases MGMT expression by modulating the acetylation of histone H3. **a** Western blot analysis of MGMT expression in TMZ-resistant and parental GBM cells. **b** Western blot analysis of MGMT expression in TMZ-resistant GBM cells transfected with LV-CRISPR-lnc-TALC. **c** Immunofluorescence analysis of MGMT and c-Met in TMZ-resistant GBM cells transfected with sh-MET. The nuclei were stained with DAPI. Scale bar = 10 μm. **d** ChIP assay of the enrichment of p300 in the MGMT promoter region normalized to IgG in LN229 and 229R GBM cells ($n = 3$). **e** Left: Immunoprecipitation of 229R and LN229 GBM cells with an antibody against p-Stat3 followed by labeling with an anti-p300 antibody. Right: Immunoprecipitation of 229R and LN229 GBM cells with an antibody against p300; the cells were subsequently labeled with the anti-p-Stat3 antibody. **f** ChIP assay of the enrichment of H3K9ac in the MGMT promoter region normalized to IgG in LN229 and 229R GBM cells ($n = 3$). **g** ChIP assay of the enrichment of p300 in the MGMT promoter region normalized to IgG in 229R GBM cells transfected with LV-CRISPR-lnc-TALC and si-Stat3 ($n = 3$). **h** Left: Immunoprecipitation of 229R GBM cells after knocking down lnc-TALC with an antibody against p-Stat3 followed by labeling with an anti-p300 antibody. Right: Immunoprecipitation of 229R GBM cells knocking down lnc-TALC with an antibody against p300 followed by labeling with an anti-p-Stat3 antibody. **i** ChIP assay of the enrichment of H3K9ac in the MGMT promoter region normalized to IgG in 229R GBM cells transfected with LV-CRISPR-lnc-TALC and si-Stat3 ($n = 3$). Data are presented as the mean ± S.D. $P$ value was determined by Student's $t$-test. Significant results were presented as NS non-significant, $*P < 0.05$, $**P < 0.01$, or $***P < 0.001$

obvious differential expression of these miRNAs between parental cells and TMZ-resistant cells using qRT-PCR (Supplementary Fig. 7c). Since Stat3-mediated regulation of MGMT does not depend on its transcriptional activity[32], we investigated whether Stat3 could specifically modify histones in the promoter region of the MGMT gene. We found that histone acetyltransferase p300 (p300) was markedly enriched in the MGMT promoter region of TMZ-resistant cells compared with that of parental cells (Fig. 6d). Protein coimmunoprecipitation indicated that p-Stat3 could bind to p300 at a higher level in TMZ-resistant GBM cells (Fig. 6e). Moreover, there was an increased enrichment of H3K9ac, H3K27ac, and H3K36ac, but not H3K4ac, levels in the MGMT promoter region in TMZ-resistant GBM cells (Fig. 6f and Supplementary Fig. 7d). Furthermore, suppression of lnc-TALC or Stat3 in TMZ-resistant GBM cells impaired the enrichment of p300 in the MGMT promoter region (Fig. 6g), the interactivity between p300 and p-Stat3 (Fig. 6h), and the enrichment of H3K9ac, H3K27ac and H3K36ac, but not H3K4ac, in the MGMT promoter region (Fig. 6i and Supplementary Fig. 7e). Collectively, these findings demonstrate that the lnc-TALC/c-Met axis regulates MGMT expression by modulating the acetylation of H3K9/27/36 through the Stat3/p300 complex.

**Knockdown of lnc-TALC restores TMZ sensitivity in vivo**. To examine the effect of lnc-TALC on the TMZ-resistant phenotype in vivo, we first verified the TMZ sensitivity of parental GBM and TMZ-resistant GBM through establishing mice orthotopic models bearing GBM xenografts via LN229 and 229R cells. Fourteen days after GBM implantation, the mice were treated intraperitoneally with TMZ (60 mg kg$^{-1}$ day$^{-1}$ per mouse) or DMSO (0.3%) every 5 days. At the same time, the mice received bioluminescence tomography every 7 days (Fig. 7a). Bioluminescent imaging revealed that the antitumor effect of TMZ was limited in TMZ-resistant GBMs (Fig. 7b, c). Mice with TMZ-resistant GBMs exhibited significantly poorer survival (Fig. 7d). Next, we evaluated the therapeutic value of lnc-TALC inhibition on TMZ-resistant GBM in vivo (Fig. 7e). Bioluminescent imaging revealed that knockdown of lnc-TALC efficiently restored the sensitivity of TMZ-resistant xenografts to TMZ treatment (Fig. 7f). The mice receiving a combined treatment demonstrated a much smaller tumor volume than the other mice (Fig. 7g) and had a significantly prolonged lifespan (Fig. 7h). Additionally, the TMZ-resistant mice showed increased levels of p-AKT, p-MAPK, p-Stat3, and MGMT (Fig. 7i). After knockdown of lnc-TALC, the p-AKT, p-MAPK, p-Stat3, and MGMT levels decreased, as shown by immunohistochemistry (Fig. 7j). Overall, these data demonstrate that lnc-TALC serves as a potential therapeutic target to overcome TMZ resistance, enhancing the benefits of TMZ therapy.

**Lnc-TALC is responsible for TMZ resistance in clinic**. Multiple samples (labeled 1, 2, and 3) from the same primary GBM patient receiving standard TMZ therapy showed the expression of lnc-TALC by qRT-PCR and in situ hybridization (ISH) assays, and subsequently, we found that a higher lnc-TALC expression showed a more obvious tendency toward recurrent neoplasms in magnetic resonance (MR) images (Fig. 8a, b), suggesting that lnc-TALC might play an important role in promoting recurrence in GBM. The expression of lnc-TALC was higher in recurrent GBM samples after TMZ treatment than in primary GBM samples (Fig. 8c). According to the Chinese Glioma Genome Atlas (CGGA) database and the qRT-PCR results, we found that MET expression levels were higher in recurrent GBM samples than in primary GBM samples (Supplementary Fig. 8a, b). In primary and paired recurrent GBM samples, immunohistochemistry

(IHC) staining detected that c-Met was highly expressed in recurrent GBM samples (Fig. 8d and Supplementary Table 3). Western blot results also showed that recurrent GBM tissues had higher levels of c-Met and p-Met than primary GBM tissues (Fig. 8e). We found a significantly positive correlation between MET and lnc-TALC expression in GBM tissues (r = 0.8688, p < 0.001, Fig. 8f). In the TCGA database, MGMT expression levels were higher in recurrent GBM samples (Supplementary Fig. 8c), and there was a positive relationship between the MET and MGMT expression levels (Supplementary Fig. 8d). Patients with low lnc-TALC expression had a significant benefit from TMZ chemotherapy (Fig. 8g, p = 0.0085) compared with patients with high lnc-TALC expression (Supplementary Table 4, p = 0.2566). The Cox regression analysis revealed that TMZ chemotherapy was associated with the overall survival of patients with low lnc-TALC expression (p = 0.027), after adjusting for age at diagnosis, age, KPS, and radiotherapy (Supplementary Table 5).

**Discussion**

LncRNAs take part in several cancer-associated processes, including miRNA silencing, epigenetic regulation, DNA damage, cell cycle control, and signal transduction pathways[37]. The current standard therapy for GBM patients consists of surgical resection followed by concurrent radiotherapy and chemotherapy. Recurrent GBM patients who develop resistance to TMZ have limited therapeutic options in the clinic. Effective treatment options for GBM, especially for TMZ resistance, are not well established. Elucidating the molecular basis of TMZ resistance could contribute to the development of logically designed combination therapies to block resistance in TMZ chemotherapy. At present, lncRNA-based TMZ-resistant mechanisms in GBM represent crucial nodal points for therapeutic intervention. In the context of GBM chemotherapy, for example, lncRNA MALAT1 has been reported to promote TMZ resistance in GBM[15,16]. However, the aberrance of lncRNA expressive profiling in TMZ-resistant GBM cells requires further exploration. LncRNA microarrays of TMZ-resistant GBM cells allows us to better understand the role of key lncRNAs in TMZ resistance.

In this study, we established TMZ-resistant GBM cells and detected alterations in lncRNA expression by microarray. We confirmed that the lncRNA MALAT1 had a high expression level in the TMZ-resistant GBM cells (data not shown) and found an unreported lncRNA in GBM, namely, lnc-TALC, that is highly expressed in TMZ-resistant GBM through comparing the expression differences in the lncRNA microarray and the TMZ IC50 after RNAi. Lnc-TALC was located on the AL358975 locus and composed of two exons with a full length of 418 nt. There are several papers published for lncARSR, a lncRNA from the same locus, composed of four exons with a full length of 591 nt[38–44]. LncARSR is upregulated in hepatocellular carcinoma (HCC), associated with tumor size and advanced stage, which directly binds to PTEN mRNA, promotes PTEN mRNA degradation, regulates PI3K/Akt pathway[38], and induces dedifferentiation of cancer stem cells by targeting STAT3 signaling in HCC cells[39]. LncARSR increases SREBP-1c and SREBP-2 expression, involved in the sterol biosynthesis after activation of PI3K/Akt pathway[40,41]. LncARSR could bind YAP to impede LATS1-induced YAP phosphorylation and facilitate YAP nuclear translocation[42], and could be incorporated into exosomes and transmitted to sensitive cells, thus disseminating sunitinib resistance in RCC[43]. In ovarian cancer, lncARSR interacts with HuR, upregulates β-catenin expression, and then activates Wnt/β-catenin signaling pathway[44].

In the present study, we observed that patients with low lnc-TALC expression exhibited a dramatic improvement in prognosis

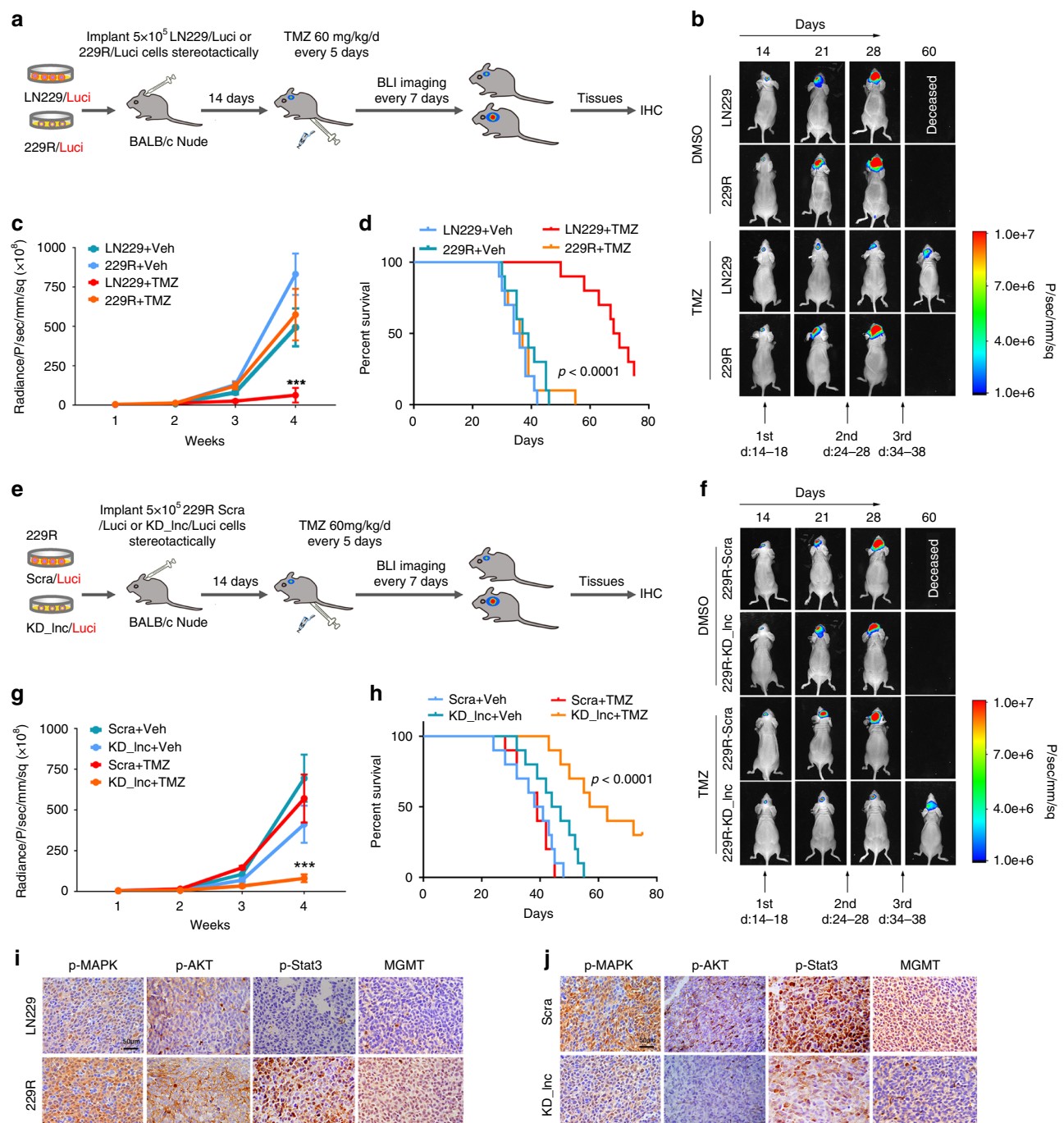

**Fig. 7** Knockdown of lnc-TALC restores TMZ sensitivity in TMZ-resistant GBM xenografts. **a** Nude mice were orthotopically xenografted with LN229 and 229R GBM cells ($5 \times 10^5$ cells) and treated intraperitoneally with TMZ (60 mg kg$^{-1}$ day$^{-1}$ per mouse) or DMSO (0.3%) every 5 days. **b** Bioluminescent images of nude mice. **c** Quantification of bioluminescent imaging signal intensities in nude mice. **d** Kaplan-Meier survival curve of nude mice is shown. **e** Nude mice were orthotopically xenografted with 229R GBM cells ($5 \times 10^5$ cells) transfected with LV-CRISPR-lnc-TALC or LV-scramble and treated intraperitoneally with TMZ (60 mg kg$^{-1}$ day$^{-1}$ per mouse) or DMSO (0.3%) every 5 days. **f** Bioluminescent images of nude mice. **g** Quantification of bioluminescent imaging signal intensities in nude mice. **h** Kaplan-Meier survival curve of nude mice is shown. **i** IHC staining of p-MAPK, p-AKT, p-Stat3 and MGMT in consecutive brain sections of mice orthotopically xenografted with 229R and LN229 GBM cells. Scale bar = 50 μm. **j** IHC staining of p-MAPK, p-AKT, p-Stat3 and MGMT in consecutive brain sections of mice orthotopically xenografted with 229R GBM cells transfected with LV-CRISPR-lnc-TALC and LV-Scra. Scale bar = 50 μm. Data are presented as the mean ± S.D. *P* value was determined by one-way ANOVA or Log-rank test. Significant results were presented as NS non-significant, *P < 0.05, **P < 0.01, or ***P < 0.001

after receiving TMZ. The knockdown of *lnc-TALC* by the CRISPR-Cas9 system resensitized resistant GBM cells to TMZ, in both in vitro and in vivo models. We failed to observe any effect of copy number variants and DNA methylation on *lnc-TALC* expression. Our results revealed that *lnc-TALC* was correlated

with the phosphorylation of Stat3, AKT, and MAPK in GBM cells. Bioinformatics analysis indicated that the phosphorylated AKT was associated with TMZ resistance in the TCGA GBM RPPA database. Recurrent gliomas from patients treated with TMZ harbor genetic alterations involved in AKT

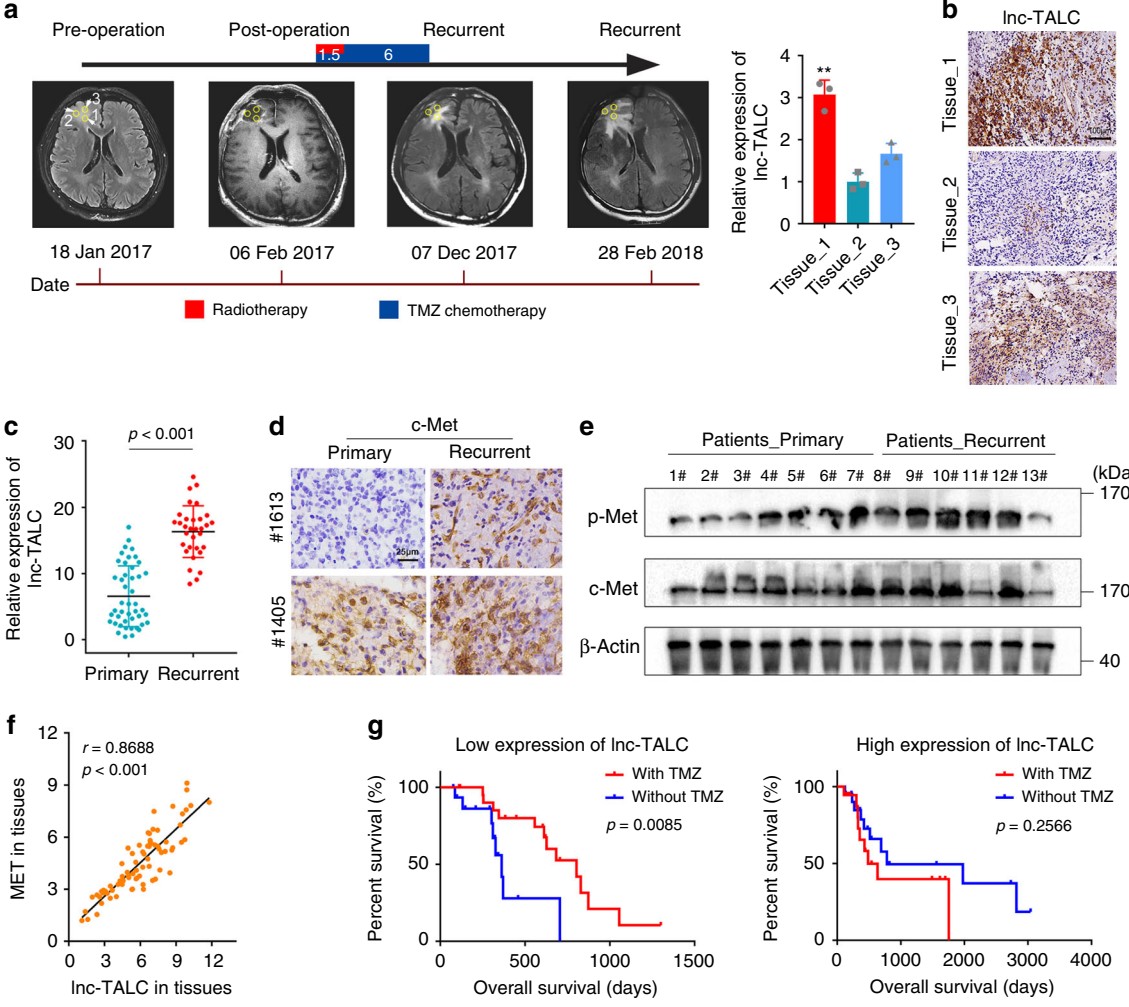

**Fig. 8** Lnc-TALC is responsible for TMZ resistance and the recurrence of GBMs in clinical patients. **a** Left: Representative MR images show the timeline of treatment histories (top intervals labeled in months) for patients with GBM and the schematic process of multiple sampling (labeled 1, 2, and 3). Right: qRT-PCR of lnc-TALC in multiple samples (labeled 1, 2, and 3) from the same primary GBM. **b** ISH of lnc-TALC in multiple samples (labeled 1, 2, and 3) from the same primary GBM. Scale bar = 100 μm. **c** qRT-PCR of lnc-TALC in primary and recurrent GBM tissues. The lnc-TALC expression was normalized to β-actin. **d** IHC of c-Met in tumor sections from primary and paired recurrent GBM samples. Scale bar = 25 μm. **e** Western blot analysis of c-Met and p-Met in primary and recurrent GBM sample tissues. **f** Pearson correlation analysis between the lnc-TALC and MET levels in 77 GBM sample tissues. **g** Kaplan-Meier curves of OS in GBM patients with or without TMZ chemotherapy. Left: Patients with low lnc-TALC expression. Right: Patients with high lnc-TALC expression. Data are presented as the mean ± S.D. P value was determined by Student's t-test, one-way ANOVA, Pearson's correlation test or Log-rank test. Significant results were presented as NS non-significant, *$P < 0.05$, **$P < 0.01$, or ***$P < 0.001$

hyperactivation[45]. Suppressing the AKT signaling pathway combination with TMZ synergistically induces autophagy and apoptosis in TMZ-resistant GBM cells[46].

As major direct substrates of AKT, FOXO factors are negatively regulated by AKT phosphorylation, which induces their nuclear export. FOXO factors are ubiquitinated and are subjected to degradation by the proteasome once in the cytoplasm. FOXO factors with well-recognized DNA-binding domains include FOXO1, FOXO3, FOXO4, and FOXO6[47]. The nuclear localization of FOXO proteins is essential for their transcriptional regulatory functions, which includes the control of genes involved in cell apoptosis, such as FasL[33], and genes involved in cell cycle regulation, such as cyclin D1 and D2[48]. Our results revealed that *lnc-TALC* expression was promoted by AKT through promoting transcription factor FOXO3 degradation in TMZ-resistant GBM cells.

Comparing the primary and recurrent GBM samples in the TCGA database, we found that c-Met had a higher expression level in recurrent samples. C-Met, as a receptor tyrosine kinase,

was an independent prognostic factor for TMZ chemotherapy. Low expression levels of c-Met are prognostic for and predict the benefits of TMZ chemotherapy in GBM[49]. C-Met mediated the plasticity of endothelial cells to undergo mesenchymal transformation, rendering GBM resistant to TMZ[50]. C-Met activation regulated various signaling pathways involving downstream kinases, such as AKT, Stat3, and MAPK[51]. Based on the ceRNA hypothesis, lncRNAs can elicit their biological activity through their ability to act as endogenous decoys for miRNAs, and such activity would, in turn, affect the distribution of miRNAs on their targets[6]. We performed bioinformatic analyses revealing that *miR-20b-3p* and *miR-335-3p* had the potential to target both *lnc-TALC* and the *MET* mRNA 3′ UTR. Furthermore, the luciferase reporter, RIP and MS2-RIP assays confirmed a direct interaction between *miR-20-3p* and its target sites in *lnc-TALC* and in the *MET* mRNA 3′ UTR regions. *MiR-20b* has been reported to repress APC and FZD6 to sustain the proneural phenotype. Additionally, FZD6 activates the CaMKII–TAK1–NLK pathway, thereby promoting Stat3 and NF-kB signaling, which are essential

for the mesenchymal phenotype[52]. GBM TMZ-sensitive clones had more proneural subtype characteristics, such as OLIG2, while GBM TMZ-resistant clones exhibited mesenchymal subtype features. Furthermore, *MET* shows reduced expression in TMZ-sensitive clones of GBM and is dramatically upregulated in TMZ-resistant clones[20]. Thus, our findings suggest that *lnc-TALC* regulated *MET* expression through competitive binding with *miR-20b-3p*.

MGMT is a ubiquitous DNA repair enzyme that has been highly conserved throughout evolution. MGMT is associated with resistance to alkylating agent cancer therapies[36]. A high MGMT expression level is mechanistically linked to TMZ resistance in GBM, and elevated MGMT expression and/or the lack of MGMT promoter hypermethylation in patient tumor specimens is associated with worse outcomes in GBM patients treated with TMZ. Modulation of this enzyme as a treatment target has been investigated for many years. MGMT regulation might be involved in the direct binding of specific miRNAs to the 3′untranslated region of MGMT transcripts, which could lead to decreased mRNA stability and/or reduced protein translation[53]. Distinct miRNAs that have been implicated as direct regulators of MGMT expression include miR-181b, miR-181d, miR-221, miR-767-3p, and miR-648[53–56]. However, these miRNAs showed no evident changes between recurrent and primary GBM samples or TMZ-resistant and TMZ-sensitive GBM cells. The epigenetic regulation of gene transcription has multiple dimensions, and MGMT has focused on CpG methylation[57]. In our current study, we did not demonstrate a methylation-level change between TMZ-sensitive and TMZ-resistant cells in the MGMT promoter regions. In addition, *lnc-TALC* overexpression did not alter the promoter methylation standard of MGMT.

In previous research, it has been well-recognized that Stat3 activity is strongly linked to TMZ resistance in GBM. Inhibition of Stat3, downstream of c-Met, overcomes TMZ resistance in GBM by downregulating MGMT expression[32]. The same results were found in our experiments. Although increasing the activity of Stat3 induces MGMT expression, there is no direct evidence indicating that Stat3-mediated regulation of MGMT depends on its transcriptional activity. MGMT expression is also regulated by chromatin remodeling[58]. Our results revealed elevated acetylation of histone H3 (H3K9ac, H3K27ac, and H3K36ac) and increased recruitment of p300 within the MGMT promoter regions of GBM cells overexpressing *lnc-TALC*, which depended on Stat3.

In conclusion, based on the lncRNA microarray, we found that lncRNA *lnc-TALC* induced by AKT mediated TMZ resistance in GBM, which trapped *miR-20b-3p*, activated c-Met and increased MGMT expression by remodeling the acetylation of histone H3 in the MGMT promoter regions. *Lnc-TALC* could serve as a therapeutic target to overcome TMZ resistance, enhancing the clinical benefits of TMZ chemotherapy in GBM patients (Fig. 9).

## Methods

**Cell lines and cell culture**. Human GBM LN229 and U251 cells were purchased from the Chinese Academy of Sciences Cell Bank. These cells were authenticated using STR assay (Genetic Testing Biotechnology, Jiangsu, China). Patient-derived GBM cells 551 W and HG7 were isolated from discarded GBM specimens. Briefly, mechanically minced tissues were digested with 0.1% trypsin (Invitrogen, USA) and 10 U ml$^{-1}$ of DNase I (Promega, USA) at 37 °C for 45 min. ACK lysis buffer (Beyotime, Shanghai, China) was used to lyse the red blood cells. After being washed, the tissues were triturated by pipetting and passed through a 100 μm cell strainer. TMZ-resistant GBM 229R, 251R, 551WR, and HG7R cells were induced from LN229, U251, 551 W and HG7 cells, respectively. All cells were cultured in Dulbecco's modified Eagle's medium (DMEM) or DMEM/F12 with 10% fetal bovine serum (Gibco, USA) at 37 °C in a humidified atmosphere with 5% $CO_2$ and were tested negative for mycoplasma contamination.

**Establishment of TMZ-resistant cells**. LN229, U251, 551 W, and HG7 cells were seeded into 96-well plates at 6000 cells per well, and IC50 of TMZ was determined.

TMZ was then added to the cell culture medium at an IC50 1/50 concentration (LN229/0.83 ± 0.26 μM, U251/1.1 ± 0.19 μM, 551 W/1.0 ± 0.22 μM, HG7/1.48 ± 0.14 μM) to culture LN229, U251, 551W, and HG7 cells in six-well plates. After the cells grew stably, the drug dose was increased in multiples. Each dose was maintained for 15 days, to the end of the fifth month. The induced TMZ-resistant cells were named 229R, 251R, 551WR, and HG7R.

**Microarray analysis**. The RNA expression profiling was performed using Agilent custom human lncRNA and mRNA microarrays (SHBIO Technology Corporation, Shanghai, China). The raw data were normalized using the quantile algorithm from the limma package in R. Heatmaps representing differentially regulated genes were generated using Cluster 3.0 and Gene Tree View. The microarray data have been deposited in NCBI's Gene Expression Omnibus (GEO) database (www.ncbi.nlm.nih.gov/geo) under accession number GSE113510. The top 10 upregulated lncRNAs of TMZ-resistant GBM cells are presented in Supplementary Table 6.

**Rapid amplification of cDNA ends (RACE)**. Total RNA was isolated with TRIzol reagent (Invitrogen, USA) according to the manufacturer's instructions. 5′RACE and 3′RACE analyses were performed with 1 μg of total RNA. The SMARTer™ RACE cDNA kit (Clontech, CA) was used according to the manufacturer's instructions. RACE PCR products were separated on a 1% agarose gel. Gel extraction products were subcloned into pGH-T vectors and sequenced bidirectionally using indicated primers (Sagene, China). The gene specific primers used for PCR are presented in Supplementary Table 7.

**Coding potential analysis**. Four different methods including Open reading frame finder from NCBI[59], phyloCSF[60], coding probability from Coding potential assessment tool (CPAT)[61], and coding potential score from coding potential calculator (CPC)[62] were performed to calculate the coding potential of lnc-TALC. Prediction of putative proteins encoded by lnc-TALC using ORF Finder. Nuclear Enriched Abundant Transcript 1 (NEAT1) served as a control non-coding gene. Glyceraldehyde-3-phosphate dehydrogenase (GAPDH) and β-actin (ACTB) served as control coding genes. We defined phyloCSF = 0[63], coding probability = 36.4%[61] and coding potential score = 0[62] as thresholds.

**Cell transfection**. Cells were transfected with siRNAs, miRNA mimics or plasmids using Lipofectamine 2000 (Invitrogen, USA). The sequences of siRNAs against specific targets are listed in Supplementary Table 8. The MET and FOXO3 plasmids were purchased from GeneChem (Shanghai, China). For knocking down lnc-TALC, LV-Cas9 lentiviruses were transfected into cells for 48 h at an MOI of 10 and selected for 7 days with puromycin at a final concentration of 3 μg ml$^{-1}$. Cells were subsequently infected with lentiviruses carrying sgRNAs designed for lnc-TALC. Twenty-four hours later, expression level of lnc-TALC was confirmed by qPCR. Lentiviruses expressing lnc-TALC, Cas9, sgRNAs targeting lnc-TALC or the negative control (LV-NC) were prepared by GeneChem. The details of the sgRNA sequences are shown in Supplementary Table 9.

**Cell apoptosis analysis**. A total of $1 \times 10^6$ cells were resuspended in a single cell suspension and washed two times with PBS solution. The cell apoptosis analysis was performed with the Annexin V-FITC or Annexin V-APC Apoptosis Detection Kit (BD Biosciences, USA) according to the manufacturer's instructions. The rates of apoptosis were detected by flow cytometry (BD FACSCanto II, USA).

**CCK-8 assay**. GBM cell viability was evaluated with the Cell Counting Kit 8 (Dojindo, Japan) and was measured at OD 450 nm with the BioTek Gen5 system (BioTek, USA).

**Colony formation assay**. Cells were seeded ($0.3 \times 10^3$ cells per well) in a six-well plate and cultured for 11 days. The resulting colonies were then washed twice with PBS and fixed with 4% formaldehyde for 10 min and stained for 30 min with 0.1% crystal violet. The number of colonies was then captured by an Olympus camera (Tokyo, Japan) and counted by ImageJ.

**5-Ethynyl-2′-deoxyuridine (EdU) assay**. Cell proliferation was detected by an EdU Cell Proliferation Assay Kit (Ribobio, Guangzhou, China) according to the manufacturer's instructions. The proportion of cells that incorporated EdU was determined with a fluorescence microscope (Nikon C2, Tokyo, Japan).

**Effect of molecule inhibitors on GBM cells**. Molecule inhibitors, including SGX-523, SAHA/NaB, decitabine, and azacitidine were used in this study. The cells were treated with SGX-523, a small-molecule inhibitor of c-Met, with a final concentration of 200 nM. SAHA and NaB, inhibitors of histone deacetylase, were added to the cells at concentrations of 2 μM and 10 mM, respectively. Decitabine and azacitidine are inhibitors of DNA methyltransferase and were added to the cells at concentrations of 0.5 μM and 10 μM, respectively.

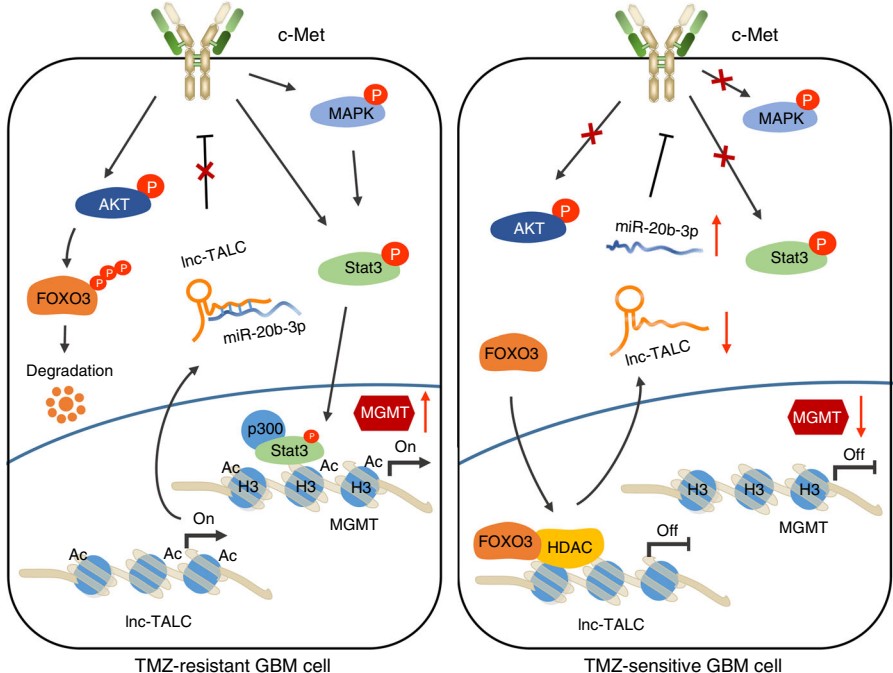

**Fig. 9** The mechanistic scheme of lncRNA lnc-TALC in GBM. The mechanistic scheme of lncRNA lnc-TALC promoting MGMT expression via regulating the c-Met signaling pathway as a competitive RNA

**Public data collection**. The CGGA microarray database was obtained from the website http://www.cgga.org.cn. TCGA microarray and protein databases were downloaded from the TCGA data portal (http://cancergenome.nih.gov/). The GSE32466 database was downloaded from the website https://www.ncbi.nlm.nih.gov/geo/query/acc.cgi?acc = GSE32466.

**Patients and specimens**. In this study, fresh GBM tissues were collected from 79 patients by surgical resection (Department of Neurosurgery, The Second Affiliated Hospital of Harbin Medical University) from 2013 to 2018. The pathological diagnoses for GBM were determined by pathologists. Recurrent GBMs were defined as an increase in residual lesions after treatment or the emergence of new lesions. The clinical characteristics of the GBM patients are presented in Supplementary Table 3–4. Written informed consent for use of the tissues and data for this research was obtained from the patients or from immediate family members or guardians. This research was approved by the Clinical Research Ethics Committee of Harbin Medical University.

**Pyrosequencing**. Genomic DNA was isolated from the LN229, 229R, 229R Scra, and 229R KD_lnc groups using a QIAamp DNA Mini Kit (Qiagen, China). The pyrosequencing analysis was carried out by Gene Tech Company Limited (Shanghai, China). Methylation values >10% in GBM cells were considered to be methylated.

**Co-Immunoprecipitation (Co-IP)**. In the coimmunoprecipitation assay, LN229, 229R, 229R Scra, and 229R KD_lnc cells were first lysed with IP lysis buffer. The lysates were incubated with 5 μg of anti-p-Stat3 or anti-p300 overnight at 4 °C. Ten microliters of protein A agarose beads were then added to the samples and incubated at 4 °C with gentle shaking for 3 h. After incubation, the bead–protein mixtures were centrifuged and washed three times with lysis buffer. The immunoprecipitated samples were further analyzed by western blot.

**Protein preparation and western blot**. Total proteins were prepared from GBM cells or clinical GBM tissues using prechilled RIPA buffer with proteinase and phosphatase inhibitor cocktails (Selleck.cn, Shanghai, China). The PVDF membranes were incubated overnight at 4 °C with primary antibodies (Supplementary Table 10) and were then incubated with an HRP-labeled secondary antibody (Zsbio Store-bio, Beijing, China) at room temperature for 1 h. The protein bands were visualized using a chemiluminescence reagent (ECL) kit (Boster, Wuhan, China). Uncropped scans of these blots are reported in Supplementary Fig. 9.

**RNA isolation and PCR**. The total RNA of the GBM cells or clinical GBM tissues was extracted using TRIzol (Invitrogen, USA) according to the manufacturer's instructions. The nuclear and cytoplasmic components were separated using 0.5% NP-40 (Solarbio, Beijing, China) with an RNAase inhibitor (Promega, USA),

followed by extraction using TRIzol reagent (Sigma, USA). One microgram of total RNA was used as a template for cDNA synthesis using a PrimeScript RT Reagent Kit (Takara, Japan). Real-time quantitative PCR was performed on triplicate samples in a reaction mix of SYBR Green (Takara, Japan) with a CFX96 Touch Real-Time PCR Detection System (Bio-Rad, USA). Quantification of the miRNA was performed with a stem-loop real-time PCR miRNA kit (RiboBio, Guangzhou, China). The levels of total RNA were normalized to β-Actin. The levels of miRNA RNA were normalized to U6 snRNA (small nuclear RNA). The expression of the indicated genes was normalized to the endogenous or exogenous reference control by using the $2^{-\Delta\Delta Ct}$ method. Sequences of the primers used for qRT-PCR in this study are listed in Supplementary Table 11. The molecular numbers of lnc-TALC, miR-20b-3p and MET were measured using the standard curve method and were calculated by relating the Ct value to the standard curve. Plasmids containing lnc-TALC or MET were purchased from GeneChem (Shanghai, China) and synthetic miR-20b-3p polynucleotide was purchased from RiboBio (Guangzhou, China).

**Luciferase reporter assay**. Cells were seeded into 96-well plates and were transfected with the GV272 luciferase vector (GeneChem, Shanghai, China). Lnc-TALC and MET wild type with potential miR-20b-3p/miR-335-3p binding sites or mutants of each binding site (Supplementary Table 12) were generated and fused to the luciferase reporter vector GV272. The firefly luciferase activity in each well was measured with a Dual Luciferase Reporter Assay System (Promega, USA) and normalized to Renilla luciferase activity, according to the manufacturer's protocol.

**Immunofluorescence**. Cells were plated on a cell smear (WHB-24-CS, Shanghai, China) in 24-well tissue culture plates. The cells were stained using standard procedures. Primary antibodies, including FOXO3, MGMT, and c-Met, were diluted in 1% BSA in PBS. After overnight incubation at 4 °C, the cells were washed three times with PBS and incubated with FITC-labeled anti-IgG antibodies (Alexa Fluor 488 and 594, Thermo Fisher) for 1 h at room temperature. The DNA was stained with DAPI (Sigma, USA) and visualized with a fluorescence microscope (Nikon C2, Tokyo, Japan) and a laser scanning confocal microscope (LSCM) (ZEISS LSM700, Germany).

**In situ hybridization**. ISH was performed with the RNA ISH Kit (BersinBio, Guangzhou, China), according to the manufacturer's instructions. Briefly, the samples were treated with pepsin for 10 min at 37 °C and incubated with 500 nM of probe at 55 °C for 4 h. The cells were subsequently incubated with 3% hydrogen peroxide to block potential endogenous peroxidase, and the probes were then detected with peroxidase-conjugated anti-DIG-Ab. Finally, the sections were stained with diaminobenzidine (DAB) for detection. The probe sequences are listed in Supplementary Table 13.

**RNA immunoprecipitation**. RIP was performed with a Magna RIP RNA-Binding Protein Immunoprecipitation Kit (Millipore, Massachusetts, USA), according to

the manufacturer's instructions. The antibodies used in the RIP assays of Ago2 were purchased from Abcam. MS2-based RIP assay with anti-GFP antibody in LN229 cells 48 h after transfection with MS2bp-YFP plasmid along with MS2bs-lnc, MS2bs-lnc Mut, or MS2bs-Rluc (GenScript, Nanjing, China). The RNA fraction precipitated by RIP was analyzed via qPCR.

**Chromatin immunoprecipitation.** ChIP assays were performed with an EZ-ChIP kit (Millipore, USA), according to the manufacturer's instructions. The ChIP-PCR products were detected with 4.8% agarose gel electrophoresis. Sequences of the primers used for ChIP-PCR in this study are listed in Supplementary Table 14. Uncropped scans of these blots are reported in Supplementary Fig. 9.

**Immunohistochemistry.** Immunohistochemistry was carried out in 4 µm paraffin sections with a three-step process and a DAB staining kit (ZSGB-BIO, Beijing, China). In brief, formalin-fixed, paraffin-embedded tissue sections were incubated at 80 °C for 15 min, dewaxed in xylene, rinsed in graded ethanol, and rehydrated in double-distilled water. For antigen retrieval, the slides were pretreated by steaming them in sodium citrate buffer for 15 min at 95 °C. After washing with PBS for 3 min, the sections were immunostained with primary antibodies to c-Met, p-MAPK, p-Stat3, p-AKT, and MGMT, and incubated at 4 °C overnight. After being washed in PBS buffer, the tissues were covered by an antimose/rabbit polymer HRP-label for 30 min. The staining reactions were performed by covering the tissue samples with the prepared DAB chromogen solution and incubating them for ~1 min to allow for proper brown color development.

**Xenograft model in vivo.** Four-week-old female athymic BALB/c nude mice were purchased from Beijing Vital River Laboratory Animal Technology Co., Ltd. (Beijing, China). A total of $3 \times 10^5$ GBM cells (LN229/229R and 229R Scra/229R KD_lnc) per mouse were stereotactically injected into the brain. After the surgery, the mice were treated for 2 weeks with DMSO or TMZ (60 mg kg$^{-1}$ day$^{-1}$). The intracranial tumors were measured with bioluminescence imaging. The mice were sacrificed, and the brain tissues were removed. The mouse brain tissue was embedded in paraffin and sectioned at a thickness of 4 µm for immunohistochemistry assays. All procedures were approved by the Committee on the Ethics of Animal Experiments of Harbin Medical University.

**Statistical analysis.** Significant differences between the groups were estimated by Student's $t$-test. One-way analysis of variance (one-way ANOVA) was used for at least three groups. The overall survival curves were used to describe the survival distributions, and the log-rank test was applied for assessing statistical significance between different groups. The survival data were further processed by using univariate and multivariate Cox regression analysis. The Pearson correlation coefficient was used to analyze the correlations between variables. GO and KEGG Pathway analysis was performed using the DAVID website (http://david.abcc.ncifcrf.gov/home.jsp). Statistically significant gene sets were visualized by Cytoscape, and GSEA was used to analyze biological processes. SsGSEA was used to calculate the enrichment score of every gene set for every sample[64]. Heatmaps were constructed and produced using Gene Cluster 3.0 and Gene Tree View software. All results are expressed as the mean ± SD. A value of $p < 0.05$ was considered to be statistically significant. All statistical analyses were performed using GraphPad software version 7.0 (GraphPad Software, CA, USA) or IBM SPSS Statistics 23.0 (SPSS, Chicago, USA).

**Study approval.** We have complied with all relevant ethical regulations for animal testing and research. All research performed was approved by the Institutional Review Board at the Second Affiliated Hospital of Harbin Medical University and was in accordance with the principles expressed at the Declaration at Helsinki. Written informed consent was received from all participants. All animal experiments were performed according to Health guidelines of Harbin Medical University Institutional Animal Use and Care Committee.

**Reporting summary.** Further information on research design is available in the Nature Research Reporting Summary linked to this article.

## Data availability

All relevant data are available from the authors. Gene expression data reported in this study have been deposited with the Gene Expression Omnibus under the accession number GSE113510. The nucleotide sequences of lnc-TALC have been deposited in the NCBI GenBank nucleotide database under the accession number MK600515. All the other data supporting the finding of this study are available within the article and its Supplementary Information files or from the corresponding author on reasonable request.

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

## Acknowledgements

This study was supported by (1) The National Key Research and Development Plan (No. 2016YFC0902502); (2). The National Natural Science Foundation of China (No. 81702972, No. 81874204, No. 81572701, No. 81772666); (3). China Postdoctoral Science Foundation (2018M640305); (4). The Research Project of the Chinese Society of Neuro-oncology, CACA (CSNO-2016-MSD12); (5). Heilongjiang Postdoctoral Science Foundation (LBH-Z18103); (6). The Research Project of the Health and Family Planning Commission of Heilongjiang Province (2017–201); and (7). The Harbin Medical University Scientific Research Innovation Fund (2017LCZX37).

## Author contributions

C.J., C.K., and J.C. conceived this study and obtained financial support. C.K., J.C. and C.J. participated in the study design and were responsible for project management. P.W., B.H., X.M., Z.L. and Y.L. performed the laboratory experiments. R.W., L.L. and C.D. collected the clinical tissue sample and data. P.W., Q.C. and J.C. co-wrote the manuscript. P.W., J.C. and Q.C. performed the data processing, statistical analysis, and bioinformatics investigations. All the authors contributed to the final manuscript.

## Additional information

**Competing interests:** The authors declare no competing interests.

