## [Peer Review File · Nature Communications]

Reviewers' comments:

Reviewer #1 (Remarks to the Author):

This manuscript investigated lncRNAs in TMZ-resistance in GBM. I have several major concerns some of which are outlined below.

1. I am very concerned with the large volume of data presented. Just to give an example, Figures 5,6,7 and 8 have more than 10-14 panels. If this paper is being revised to address additional comments/suggestions, it will continue to lack focus that is needed for a good, solid paper to move the field forward. If the authors care about this they might want to read the following article in Nature, entitled "Publish houses of brick, not mansions of straw".

2. Data lacks clarity. There are several examples of this including: (1) it is almost impossible to see the contents in Figures 1B-D, Figure 2B, Figures 4A-C, etc. (2) Figure 2A: two types of resistant cells. It is not clear from the line 327 or from the Figure or from the Figure legend, which cells were used. (3) line 318, what is ssGSEA? (4) line 332, which CRISPR/Cas9 system was used?

3. Data lacks controls. Again, there are many examples including (1) only one siRNA was used in Figure 2, (2) Figure 2C, what is the effect of kd of the lncRNA in untreated condition?

4. It's not clear what lnc-TALC is. What is the Gene symbol? Is there only one isoform? When I used the coordinates from Figure 3C, it looks like there are many isoforms? However, the authors show only one transcript.

5. What is the number of molecules of lnc-TALC, miR-20b-3p and MET mRNA in these cells?

Reviewer #3 (Remarks to the Author):

Summary: In this study, the authors showed that a long non-coding RNA (temozolomide-associated lncRNA in glioblastoma recurrence) (lnc-TALC) competed and bound to miR-20b-3p. miR-20b-3p inhibits the activation of Receptor Tyrosine Kinase, c-Met. c-Met induces the activation of Akt which in turn phosphorylates and inactivates the tumor suppressor FOXO3. c-Met also activates MAPK/Stat3/p300 pathway. Stat3/p300 complex modifies the Histone 3 in MGMT promoter and induces MGMT expression in TMZ resistant GBM cells. Methyl Guanine Methyl Transferase (MGMT) is a DNA repair enzyme that removes methyl groups on DNA. Temozolomide (TMZ) is the frontline chemotherapy drug for GBM. TMZ alkylates DNA and causes DNA damage in cancer cells which if left unrepaired, leads to apoptosis and cell death. However, activation of DNA repair enzyme, MGMT and removal of alkyl group on DNA and repair of DNA by this enzyme leads to TMZ resistant GBM. The authors suggest that because lnc-TALC regulates these pathways through miR-20b-3p, lnc-TALC can serve as a new target for therapy of TMZ-resistant GBM. Overall, this is an interesting study focusing on a novel long non-coding RNA that sheds light to a potential mechanism that explains TMZ-resistant GBM. The paper will need some editing to improve the readability. I was confused a number of times during reading of the manuscript. Specific comments:

1. The current title is not very clear. The authors may want to consider alternative that specifies miR20b-3p since that is the proposed direct target.

2. Introduction; line 45: The sentence "For example, the pancreatic cancer risk variant of LINC00673 creates a miR-1231 binding site and interferes with PTPN11 degradation"⁸. This sentence needs to be expanded a little bit more as it is a supporting evidence of importance of embedded miRNAs in long non-coding RNAs that regulate different signaling pathways. In

reference 8, the authors showed that LINC00673 is a tumor suppressor and diminishes SRC-ERK oncogenic signaling.

3. Introduction; line 66: “..... development and progression by stimulating the PI3K/AKT, Ras/MAPK, JAK/STAT, SRC, and Wnt/ β -catenin signaling pathways, among others”. No references were provided at the end of the sentence. A reference for each of these pathways should be provided.

4. Materials and Methods: Line 211: “The levels of miRNA RNA were normalized to U6”. The word “RNA” should be deleted. U6 SnRNA (small nuclear RNA) is a more accurate term.

Lines 212 to 217: It is not necessary to mention the PCR condition. Also if the quantitative real time PCR was used, why at the end the PCR products were run on a 4.8% agarose gel?

5. Results; Line 331-332, Figure 4E: The authors did not explain well how they knocked down Inc-TALC by CRISPR-Cas9.

Line 401. The sentence “The RIP assay showed that Inc-TALC and MET could bind Ago2 (Fig. 5B-D)”. This should be changed to “Inc-TALC and c-MET transcript could bind Ago2”.

Figure 5B: In this figure, the RNA and RNA binding protein should be labelled. The figure seems to depict the RIP assay in general.

Line 402; Figure 5B-D should be separated and each figure should be explained separately e.g. Figure 5B, 5C, 5D because each figure contains results that needs to be explained. In general, figures and results should be better explained.

In Figure 9, the phosphorylated FOXO3 is shown in the nucleus, and its degradation in the cytoplasm. This does not seem to be consistent with the literature which suggests that phosphorylation of FOXO3 takes place in the cytoplasm by phosphorylated AKT and subsequently the phosphorylated FOXO3 gets ubiquitinated and degraded in the cytoplasm [(References (12) and (32) in this article)].

The authors suggest that Inc-TALC can be targeted for therapy of TMZ-resistance GBM, but no such effort such as siRNA approach was shown.

Point-by-Point Response to Reviewer's Comments

Reviewer #1 (Remarks to the Author):

1. I am very concerned with the large volume of data presented. Just to give an example, Figures 5,6,7 and 8 have more than 10-14 panels. If this paper is being revised to address additional comments/suggestions, it will continue to lack focus that is needed for a good, solid paper to move the field forward. If the authors care about this they might want to read the following article in Nature, entitled "Publish houses of brick, not mansions of straw".

Response: Thank you for your professional review of our article. We have carefully read William G. Kaelin Jr.'s paper titled "Publish houses of brick, not mansions of straw"¹. This article should be compulsory reading for everyone, as it accurately reflects where we stand today in terms of publishing our research. Kaelin, a very successful researcher (awarded the 2016 Lasker prize) gave some insightful opinions. According to Kaelin's and your important suggestions, we have made the following adjustments and exhibited the more solid data to support the conclusion of this paper.

(1) The original Figure 4A and 4C panels were moved into the revised Figure S4, making it much clearer that c-Met regulated by *lnc-TALC* is highly expressed in recurrent GBM and is required for TMZ resistance.

Revised Figure 4:

(2) The original Figure 5A, 5G and 5H (right) panels were moved to the revised Figure S5. The original Figure 5C and 5D were merged into the revised Figure 5B. The original Figure 5I and 5J were merged into the revised Figure 5F. The above changes better clarify the conclusion that *lnc-TALC* competitively binds miR-20b-3p by targeting the *MET* 3'UTR region.

Revised Figure 5:

(3) The original Figure 6B, 6D and 6F (right) panels were moved to revised Figure S6, better supporting the contention that the *lnc-TALC/c-Met* axis activates the Stat3/p300 complex to increase MGMT expression by modulating acetylation of histone H3.

Revised Figure 6:

- (4) The original Figure 8E, 8F, 8J and 8K panels were moved to revised Figure S8 and the original Figure 8A and 8B were merged into the revised Figure 8A, making the conclusion that knockdown of *lnc-TALC* restores TMZ sensitivity in TMZ-resistant GBM xenografts much more intelligible.

Revised Figure 8:

Our results show that *lnc-TALC* could serve as a therapeutic target to overcome TMZ resistance, enhancing the clinical benefits of TMZ chemotherapy in GBM. Thank you for your valuable suggestions for improving the quality of our work.

- Data lacks clarity. There are several examples of this including: (1) it is almost impossible to see the contents in Figures 1B-D. Figure 2B, Figures 4A-C, etc. (2) Figure 2A: two types of resistant cells. It is not clear from the line 327 or from the Figure or from the Figure legend, which cells were used. (3) line 318, what is ssGSEA? (4) line 332, which CRISPR/Cas9 system was used?

Response: We thank you for your careful review and valuable suggestions.

- (1) We increased the font size of the original Figure 1B-D, Figure 2B and Figure 4A (the revised Figure S4A), 4B (the revised Figure 4A), 4C (the revised Figure S4B); removed some unimportant or illegible labels from the original Figure 1C and Figure 4A (the revised Figure S4A), 4B (the revised Figure 4A); and added additional labels to the original Figure 4C (the revised Figure S4B).

Revised Figure 1:

Revised Figure 2:

Revised Figure 4:

Revised Figure S4:

- (2) We added the names of resistant cells in the revised Figure 2A, the figure legend, and the manuscript. Thank you very much.

Revised Figure 2:

- (3) Single-sample GSEA (ssGSEA)², an extension of Gene Set Enrichment Analysis (GSEA)³, calculates separate enrichment scores for each pairing of a sample and gene set. Each ssGSEA enrichment score represents the degree to which genes in a particular gene set are coordinately up- or down- regulated within a sample. In this manner, ssGSEA projects a single sample's gene expression profile from the space of a single gene onto the space of gene sets. The ssGSEA projection transforms data to a higher-level (pathways instead of genes) space representing a more biologically interpretable set of features⁴. We have described this method in the methods section of the revised manuscript.
- (4) The functions of CRISPR (Clustered Regularly Interspaced Short Palindromic Repeats) and CRISPR-associated (Cas) genes are essential in adaptive immunity in select bacteria and archaea, enabling the organisms to respond to and eliminate invading genetic material⁵. The CRISPR/Cas9 system has been shown to be a powerful genome-editing tool in a variety of organisms⁶. There are three types of CRISPR/Cas systems. The type II system uses single

effector enzymes, such as Cas9, to cleave dsDNA, and the DNA double-strand cutting process can be completed only by the combination of Cas9 protein and sgRNA^{7,8,9}. Because of this unique feature, the CRISPR/Cas type II system has been extensively utilized studied as a genetic engineering tool from bacteria to mammals⁹. The CRISPR/Cas9 system has been used to successfully knock down various lncRNAs^{10, 11, 12, 13, 14}. Application of the CRISPR/Cas9 system for knocking down genes has been mastered and exemplified in our previous studies^{15, 16}. Therefore, we designed a strategy to knockdown *lnc-TALC* using two pairs of sgRNAs, and the sgRNA sequences are shown in Supplementary Table 8. SgRNAs targeting the *lnc-TALC* gene were cloned into GV504-U6-*lnc-TALC*-sgRNAs-SV40-EGFP. Full-length Cas9 was cloned into GV371-CMV-hSpCas9-SV40-Puro. For knocking down *lnc-TALC*, LV-Cas9 lentiviruses were transfected into cells for 48 h at an MOI of 10 and selected for 7 days with puromycin at a final concentration of 3 µg/ml. Cells were subsequently infected with lentiviruses carrying sgRNAs designed for *lnc-TALC*. After 24 h, expression levels of *lnc-TALC* were confirmed by qPCR. Plasmid pairs and lentiviruses of CRISPR/Cas9/*lnc-TALC*-sgRNA were synthesized and purchased from GeneChem Company (Shanghai, China). We have added the above contents into the methods section. Thank you again for your valuable suggestions.

3. Data lacks controls. Again, there are many examples including (1) only one siRNA was used in Figure 2, (2) Figure 2C, what is the effect of kd of the lncRNA in untreated condition?

Response: We appreciate your valuable suggestions. (1) We used another siRNA termed siRNA_2 (the original siRNA named as is termed siRNA_1) and presented the corresponding results in revised Figure 2A and Figure 3D-E. Please check them. Thank you. (2) *In vivo*, the effects of *lnc-TALC* knockdown in untreated samples are shown in Figure 7G-H. There was no statistically significant difference in tumor size and survival time between the scrambled and *lnc-TALC* knockdown groups in the untreated conditions. Knockdown of *lnc-TALC* in resistant GBM cells did not significantly promote apoptosis or inhibit cell colony formation or proliferation (Figure 2D-F). In addition, we conducted CCK-8 assays to assess cell viability in lncRNA knockdown groups in 229R/HG7R/251R/551WR cells in untreated conditions. However, there was no statistically significant difference between the scrambled

and *lnc-TALC* knockdown groups in the untreated condition (Responding Figure 1).

Revised Figure 2:

Revised Figure 3:

Revised Figure 3:

Responding Figure 1:

4. Its not clear what *lnc-TALC* is. What is the Gene symbol? Is there only one isoform? When I used the coordinates from Figure 3C, it looks like there are many isoforms? However, the authors show only one transcript.

Response: Thank you very much for your review of our work. In this study, the top 10 up-regulated lncRNAs in TMZ-resistant GBM cells were identified to elucidate the underlying lncRNA-based mechanisms of TMZ resistance. We found that *lnc-TALC* was both highly expressed and associated with TMZ resistance in GBM cells through RNAi and qPCR assays. *lnc-TALC* is a long non-coding RNA in Agilent custom human lncRNA microarrays (Agilent-74348, Biotechnology Corporation, Shanghai, China). In the Ensemble database, this long non-coding RNA is named AL358975.1-201, a transcript of the gene ENSG00000233086, which has four isoforms. AL358975.1-201 is one of the top10 lncRNAs

in TMZ-resistant GBM cells compared to parental cells. LncRNA microarrays (Agilent-74348) contain the AL358975.1-201, AL358975.1-203, and AL358975.1-204 isoforms. In the lncRNA microarray, there was no significant difference between AL358975.1-203 and AL358975.1-204 in the LN229 and 229R cell lines (Responding Figure 2A). qPCR revealed that expression levels of AL358975.1-202 were also not different (Responding Figure 2B). Thus, AL358975.1-202, AL358975.1-203, and AL358975.1-204 do not fit our study.

Responding Figure 2:

5. What is the number of molecules of lnc-TALC, miR-20-3p and MET mRNA in these cells?

Response: Thank you for your valuable suggestion. The lengths of *lnc-TALC*, *miR-20-3p* and *MET* mRNA are 328bp, 22bp and 6566bp, respectively. The number of molecules of *lnc-TALC*, *miR-20-3p* and *MET* mRNA was detected in LN229/229R, U251/251R, 551W/551WR and HG7/HG7R by qPCR (Responding Figure 3). Thank you again for your review.

Responding Figure 3:

Reviewer #3 (Remarks to the Author):

1. The current title is not very clear. The authors may want to consider alternative that specifies miR020b-3p since that is the proposed direct target.

Response: Thank you for the constructive comments and suggestions. According to your advice, we have created a new title, “***Lnc-TALC* promotes O⁶-methylguanine-DNA methyltransferase expression via regulating the c-Met pathway by competitively binding with *miR-20b-3p***” instead of “LncRNA *lnc-TALC* promotes O⁶-methylguanine-DNA methyltransferase expression via regulating the c-Met signaling pathway as a competitive RNA”. This updated title conveys more clearly that *miR-20b-3p* serves as the underlying direct target of *lnc-TALC* and the c-Met transcript. Thank you again.

2. Introduction; line 45: The sentence “For example, the pancreatic cancer risk variant of LINC00673 creates a miR-1231 binding site and interferes with PTPN11 degradation”⁸. This sentence needs to be expanded a little bit more as it is a supporting evidence of importance of embedded miRNAs in long non-coding RNAs that regulate different signaling pathways. In reference 8, the authors showed that LINC00673 is a tumor suppressor and diminishes SRC-ERK oncogenic signaling.

Response: We appreciate your valuable suggestion to improve our work. LncRNAs have been described as pseudogenes that compete for miRNA binding, playing widespread roles in gene regulation and cellular processes^{17, 18}. For example, *LINC00673* acts as a tumor suppressor, diminishing SRC-ERK oncogenic signaling. However, a G>A change at rs1111655237 in exon 4 of *LINC00673* creates a target site for miR-1231 binding, decreases PTPN11 ubiquitination, and attenuates the effect of *LINC00673* in an allele-specific manner, conferring susceptibility to tumorigenesis¹⁹ and indicating the importance of embedded miRNAs in lncRNAs regulating oncogenic signaling pathways. Emerging evidence has revealed that lncRNAs, as competitive RNAs^{18, 20}, mediate postoperative treatment resistance in some cancers^{21, 22}. *Lnc-RI*, a radiation-inducible lncRNA molecule involved in the radiation-induced DNA damage response, acts as a ceRNA to stabilize *RAD51* mRNA via competitively binding with *miR-193a-3p* and releasing of its inhibition on *RAD51* expression²². Thus, the transcriptome profiling alteration of lncRNAs still needs to be illustrated in resistant tumor cells. According to your comments, we have adjusted the introduction and marked changes in the revised manuscript. Thank you again for your work and splendid suggestions.

3. Introduction; line 66: “..... development and progression by stimulating the PI3K/AKT, Ras/MAPK, JAK/STAT, SRC, and Wnt/ β -catenin signaling pathways, among others”. No references were provided at the end of the sentence. A reference for each of these pathways should be provided.

Response: We appreciate your comment. According to your suggestion, we have added references to each of these pathways that are closely related to c-Met, including the PI3K/AKT, Ras/MAPK, JAK/STAT, SRC, and Wnt/ β -catenin signaling pathways. References have been added to the sentence “In cancer cells, aberrant c-Met axis activation, closely related to c-Met gene mutations, overexpression, and amplification, promotes tumor development and progression by stimulating the PI3K/AKT²³, Ras/MAPK²⁴, JAK/STAT²⁵, SRC²⁶, and Wnt/ β -catenin²⁷ signaling pathways, among others^{28, 29}”. Thank you again for the valuable suggestion to improve our manuscript.

4. Materials and Methods: Line 211: “The levels of miRNA RNA were normalized to U6”. The word “RNA” should be deleted. U6 SnRNA (small nuclear RNA) is a more accurate term.

Lines 212 to 217: It is not necessary to mention the PCR condition. Also if the quantitative real time PCR was used, why at the end the PCR products were run on a 4.8% agarose gel?

Response: Thank you for these suggestions. We have deleted the word “RNA” in the sentence “The levels of miRNA RNA were normalized to U6”. U6 SnRNA (small nuclear RNA) has been used in the revised manuscript instead of “U6”. We regret our vague description in the methods. PCR products derived from CHIP-PCR were run on a 4.8% agarose gel to quantitatively compare differences among groups based on the gray scale. We have revised the description of the experimental method in the revised manuscript.

5. Results; Line 331-332, Figure 4E: The authors did not explain well how they knocked down Inc-TALC by CRISPR-Cas9.

Response: We appreciate your comment. The CRISPR/Cas9 system has been declared to successfully knock down various lncRNAs^{10, 11, 12, 13, 14}. Application of the CRISPR/Cas9

system for knocking down genes has been mastered and validated in our previous studies^{15,16}. Therefore, we designed a strategy to knock-down *lnc-TALC* using two pairs of sgRNAs, and the sgRNA sequences are shown in Supplementary Table 8. SgRNAs targeting the *lnc-TALC* gene were cloned into GV504-U6-*lnc-TALC*-sgRNAs-SV40-EGFP. Full-length Cas9 was cloned into GV371-CMV-hSpCas9-SV40-Puro. For knocking down *lnc-TALC*, LV-Cas9 lentiviruses were transfected into cells for 48 h at an MOI of 10 and selected for 7 days with puromycin at a final concentration of 3 µg/ml. Cells were subsequently infected with lentiviruses carrying sgRNAs designed for *lnc-TALC*. Twenty-four hours later, expression level of *lnc-TALC* was confirmed by qPCR. The plasmid pairs and lentiviruses for CRISPR/Cas9/*lnc-TALC*-sgRNA were synthesized and purchased from GeneChem Company (Shanghai, China). We added the above content to the methods section. Thank you again for your valuable suggestions.

Minor comments:

1. Line 401. The sentence “The RIP assay showed that *lnc-TALC* and MET could bind Ago2 (Fig. 5B-D)”. This should be changed to “.....*lnc-TALC* and c-MET transcript could bind Ago2”.

Response: Thank you for your suggestion. We have changed the sentence to “The RIP assay showed that *lnc-TALC* and c-MET transcript could bind Ago2 (Fig. 5B-D)” instead of “The RIP assay showed that *lnc-TALC* and MET could bind Ago2 (Fig. 5B-D)”. Thank you again for your careful work.

2. Figure 5B: In this figure, the RNA and RNA binding protein should be labelled. The figure seems to depict the RIP assay in general.

Response: Thank you for your suggestion. We have labelled the RNA and RNA binding protein in the revised Figure 5B. A RIP assay was performed as reported previously by our team¹⁵. The supernatant of cell lysate was incubated with protein A/G magnetic beads coated with anti-Ago2 antibody. The beads were then washed, and RNA was isolated and processed for PCR analysis of *lnc-TALC* and c-MET transcript. Thank you again for your precise suggestion.

3. Line 402; Figure 5B-D should be separated and each figure should be explained separately e.g. Figure 5B, 5C, 5D because each figure contains results that needs to be explained. In general, figures and results should be better explained.

Response: We thank you for your careful review. The legends and results of the original Figure 5B-D (the revised Figure 5A-B) have been separately described and further explained in the revised manuscript. The revised Figure 5A is a schematic of the RIP-PCR assay. Supernatants of cell lysate were incubated with protein A/G magnetic beads coated with anti-Ago2 antibody. Beads were then washed, and RNA was isolated and processed for PCR analysis of *lnc-TALC* and the c-MET transcript. The revised Figure 5B (upper) shows enrichment of *lnc-TALC* and c-MET transcript with Ago2 normalized to IgG in LN229 cells transfected with LV-scramble or LV-*lnc-TALC*. We overexpressed *lnc-TALC* in LN229 cells and observed increasing enrichment of *lnc-TALC* and decreasing enrichment of c-MET transcript in Ago2 compared to controls. The revised Figure 5B (lower) shows enrichment of *lnc-TALC* and c-MET transcript with Ago2 normalized to IgG in 229R cells with knock-down scramble or *lnc-TALC*. We knocked down *lnc-TALC* in 229R cells and observed decreasing enrichment of *lnc-TALC* and increasing enrichment of c-MET transcript in Ago2 compared to controls. We have added the above content in the results and figure legends. Thank you again for your careful review of our work.

4. In Figure 9, the phosphorylated FOXO3 is shown in the nucleus, and its degradation in the cytoplasm. This does not seem to be consistent with the literature which suggests that phosphorylation of FOXO3 takes place in the cytoplasm by phosphorylated AKT and subsequently the phosphorylated FOXO3 gets ubiquitinated and degraded in the cytoplasm [(References (12) and (32) in this article)].

Response: Thank you for your crucial suggestions. We have carefully checked the references for phosphorylated FOXO3 induction by activated AKT, which is ubiquitinated and degraded in the cytoplasm^{21, 30}. We have revised Figure 9 according to the above biological process. We thank you again for your careful and critical comments.

Revised Figure 9:

5. The authors suggest that *lnc-TALC* can be targeted for therapy of TMZ-resistance GBM, but no such effort such as siRNA approach was shown.

Response: We thank you for your suggestion. In the present study, we used the siRNA to knock-down the 10 selected lncRNAs and validated the RNA expressive level (Fig. S2B) for further loss-of-function analysis in TMZ-resistant GBM cells. Knockdown of *lnc-TALC* inhibited TMZ-resistance in two types of resistant cells (Fig. 2A). In addition, we stably knocked down *lnc-TALC* in TMZ-resistant cells using the CRISPR-Cas9 system (Fig. S2C). Knockdown of *lnc-TALC* in resistant GBM cells significantly decreased cell viability, promoted apoptosis, and inhibited cell colony formation and proliferation in response to TMZ treatment (Fig. 2C-F). We also evaluated the therapeutic value of *lnc-TALC* *in vivo* (Fig. 7E). Bioluminescent imaging revealed that knockdown of *lnc-TALC* efficiently restored the sensitivity of TMZ-resistant xenografts to TMZ treatment (Fig. 7F). Mice receiving a combined treatment exhibited much smaller tumor volume than others (Fig. 7G) and exhibited a significantly prolonged lifespan (Fig. 7H). Thank you again for your valuable suggestions.

References:

1. Kaelin WG, Jr. Publish houses of brick, not mansions of straw. *Nature* 2017, **545**(7655): 387.

2. Barbie DA, Tamayo P, Boehm JS, Kim SY, Moody SE, Dunn IF, *et al.* Systematic RNA interference reveals that oncogenic KRAS-driven cancers require TBK1. *Nature* 2009, **462**(7269): 108-112.
3. Subramanian A, Tamayo P, Mootha VK, Mukherjee S, Ebert BL, Gillette MA, *et al.* Gene set enrichment analysis: a knowledge-based approach for interpreting genome-wide expression profiles. *Proc Natl Acad Sci U S A* 2005, **102**(43): 15545-15550.
4. Guttman M, Amit I, Garber M, French C, Lin MF, Feldser D, *et al.* Chromatin signature reveals over a thousand highly conserved large non-coding RNAs in mammals. *Nature* 2009, **458**(7235): 223-227.
5. Barrangou R, Fremaux C, Deveau H, Richards M, Boyaval P, Moineau S, *et al.* CRISPR provides acquired resistance against viruses in prokaryotes. *Science* 2007, **315**(5819): 1709-1712.
6. Charpentier E, Doudna JA. Biotechnology: Rewriting a genome. *Nature* 2013, **495**(7439): 50-51.
7. Bhaya D, Davison M, Barrangou R. CRISPR-Cas systems in bacteria and archaea: versatile small RNAs for adaptive defense and regulation. *Annual review of genetics* 2011, **45**: 273-297.
8. Wiedenheft B, Sternberg SH, Doudna JA. RNA-guided genetic silencing systems in bacteria and archaea. *Nature* 2012, **482**(7385): 331-338.
9. Ho T-T, Zhou N, Huang J, Koirala P, Xu M, Fung R, *et al.* Targeting non-coding RNAs with the CRISPR/Cas9 system in human cell lines. *Nucleic Acids Research* 2015, **43**(3): e17-e17.
10. Liu J, Ben Q, Lu E, He X, Yang X, Ma J, *et al.* Long noncoding RNA PANDAR blocks CDKN1A gene transcription by competitive interaction with p53 protein in gastric cancer. *Cell death & disease* 2018, **9**(2): 168.
11. Xing Z, Zhang Y, Liang K, Yan L, Xiang Y, Li C, *et al.* Expression of Long Noncoding RNA YIYA Promotes Glycolysis in Breast Cancer. *Cancer research* 2018, **78**(16): 4524-4532.
12. Peng WX, Huang JG, Yang L, Gong AH, Mo YY. Linc-RoR promotes MAPK/ERK signaling and confers estrogen-independent growth of breast cancer. *Molecular cancer* 2017, **16**(1): 161.
13. Ho TT, Zhou N, Huang J, Koirala P, Xu M, Fung R, *et al.* Targeting non-coding RNAs with the CRISPR/Cas9 system in human cell lines. *Nucleic Acids Res* 2015, **43**(3): e17.
14. Aparicio-Prat E, Arnan C, Sala I, Bosch N, Guigo R, Johnson R. DECKO: Single-oligo, dual-CRISPR deletion of genomic elements including long non-coding RNAs. *BMC genomics* 2015, **16**: 846.

15. Chen Q, Cai J, Wang Q, Wang Y, Liu M, Yang J, *et al.* Long Noncoding RNA NEAT1, Regulated by the EGFR Pathway, Contributes to Glioblastoma Progression Through the WNT/beta-Catenin Pathway by Scaffolding EZH2. *Clinical cancer research : an official journal of the American Association for Cancer Research* 2017.
16. Han B, Cai J, Gao W, Meng X, Gao F, Wu P, *et al.* Loss of ATRX suppresses ATM dependent DNA damage repair by modulating H3K9me3 to enhance temozolomide sensitivity in glioma. *Cancer Letters* 2018, **419**: 280-290.
17. Ulitsky I, Bartel DP. lincRNAs: genomics, evolution, and mechanisms. *Cell* 2013, **154**(1): 26-46.
18. Salmena L, Poliseno L, Tay Y, Kats L, Pandolfi PP. A ceRNA hypothesis: the Rosetta Stone of a hidden RNA language? *Cell* 2011, **146**(3): 353-358.
19. Zheng J, Huang X, Tan W, Yu D, Du Z, Chang J, *et al.* Pancreatic cancer risk variant in LINC00673 creates a miR-1231 binding site and interferes with PTPN11 degradation. *Nature genetics* 2016, **48**(7): 747-757.
20. Karreth FA, Pandolfi PP. ceRNA cross-talk in cancer: when ce-bling rivalries go awry. *Cancer discovery* 2013, **3**(10): 1113-1121.
21. Qu L, Ding J, Chen C, Wu ZJ, Liu B, Gao Y, *et al.* Exosome-Transmitted lncARSR Promotes Sunitinib Resistance in Renal Cancer by Acting as a Competing Endogenous RNA. *Cancer Cell* 2016, **29**(5): 653-668.
22. Shen L, Wang Q, Liu R, Chen Z, Zhang X, Zhou P, *et al.* LncRNA lnc-RI regulates homologous recombination repair of DNA double-strand breaks by stabilizing RAD51 mRNA as a competitive endogenous RNA. *Nucleic acids research* 2018, **46**(2): 717-729.
23. Yao Y, Dou C, Lu Z, Zheng X, Liu Q. MACC1 suppresses cell apoptosis in hepatocellular carcinoma by targeting the HGF/c-MET/AKT pathway. *Cellular physiology and biochemistry : international journal of experimental cellular physiology, biochemistry, and pharmacology* 2015, **35**(3): 983-996.
24. Wang J, Gui Z, Deng L, Sun M, Guo R, Zhang W, *et al.* c-Met upregulates aquaporin 3 expression in human gastric carcinoma cells via the ERK signalling pathway. *Cancer letters* 2012, **319**(1): 109-117.
25. Ding X, Ji J, Jiang J, Cai Q, Wang C, Shi M, *et al.* HGF-mediated crosstalk between cancer-associated fibroblasts and MET-unamplified gastric cancer cells activates coordinated tumorigenesis and metastasis. *Cell death & disease* 2018, **9**(9): 867.
26. Ponzetto C, Bardelli A, Zhen Z, Maina F, dalla Zonca P, Giordano S, *et al.* A multifunctional

docking site mediates signaling and transformation by the hepatocyte growth factor/scatter factor receptor family. *Cell* 1994, **77**(2): 261-271.

27. Kim KH, Seol HJ, Kim EH, Rhee J, Jin HJ, Lee Y, *et al.* Wnt/beta-catenin signaling is a key downstream mediator of MET signaling in glioblastoma stem cells. *Neuro-oncology* 2013, **15**(2): 161-171.
28. Zhang Y, Xia M, Jin K, Wang S, Wei H, Fan C, *et al.* Function of the c-Met receptor tyrosine kinase in carcinogenesis and associated therapeutic opportunities. *Molecular cancer* 2018, **17**(1): 45.
29. Organ SL, Tsao MS. An overview of the c-MET signaling pathway. *Therapeutic advances in medical oncology* 2011, **3**(1 Suppl): S7-S19.
30. Brunet A, Bonni A, Zigmond MJ, Lin MZ, Juo P, Hu LS, *et al.* Akt promotes cell survival by phosphorylating and inhibiting a Forkhead transcription factor. *Cell* 1999, **96**(6): 857-868.

Reviewers' comments:

Reviewer #3 (Remarks to the Author):

The authors did a good job in addressing my comments.

Reviewer #4 (Remarks to the Author):

The authors have addressed some of the concerns raised by the previous reviewer #1. However, two major issues have not been addressed and must be experimentally clarified prior to publication:

Gene Name and Sequence Identity: According to the UCSC Genome Browser, there is a transcript annotated in the area that the authors mention that has been named LNCARSR. There, it also seems to be an absolutely testis-specific transcript. This should be mentioned in the text. Additionally, there are seven papers in PubMed listed for IncARSR, so the authors need to compare the published transcript to their gene and discuss functional differences and overlap with this transcript from the AL358975 locus. For example, there are three previous papers stating that IncARSR affects AKT signaling. The precise sequence and transcript identifier (e.g. accession number ENST...) must be given for the sequence used here in this study. A proper workflow would be to perform a RACE experiment (5' and 3') for the transcripts in this locus and then quantify the detected transcripts specifically by RT-qPCR in the sensitive and resistant cell lines. These data also needs to be included into the manuscript and the used primers and siRNAs must be accordingly mapped to the sequence. These sequences then also need to be evaluated for their coding potential (at least in silico) and compared to all annotated (and at least in part already published and named) transcript from this locus.

Number of molecules: The relative expression of ceRNA, target and microRNA is an essential point to propose such a mechanism. The authors have not adequately responded to this previous concern as it is not possible to extract molecule numbers per cell from the data provided. The authors need to establish the number of molecules per cell for all three RNAs: Inc-TALC, miR-20b-3p and c-MET. Therefore, they need to perform a dilution series with a plasmid or synthetic polynucleotide for qPCR and then compare the signals from a defined number of cells to estimate the number of molecules per cell.

Point-by-Point Response to Reviewer's Comments

Reviewer #3 (Remarks to the Author):

1. Gene Name and Sequence Identity: According to the UCSC Genome Browser, there is a transcript annotated in the area that the authors mention that has been named LNCARSR. There, it also seems to be an absolutely testis-specific transcript. This should be mentioned in the text. Additionally, there are seven papers in PubMed listed for lncARSR, so the authors need to compare the published transcript to their gene and discuss functional differences and overlap with this transcript from the AL358975 locus. For example, there are three previous papers stating that lncARSR affects AKT signaling. The precise sequence and transcript identifier (e.g. accession number ENST...) must be given for the sequence used here in this study.

A proper workflow would be to perform a RACE experiment (5' and 3') for the transcripts in this locus and then quantify the detected transcripts specifically by RT-qPCR in the sensitive and resistant cell lines. These data also needs to be included into the manuscript and the used primers and siRNAs must be accordingly mapped to the sequence. These sequences then also need to be evaluated for their coding potential (at least in silico) and compared to all annotated (and at least in part already published and named) transcript from this locus.

Response: We thank you for your careful and important suggestions.

- (1) Thank you for your reminding that the UCSC Genome Browser updates the annotations for this gene and names it as “lncARSR”. Although RNA seq expression profile in the NONCODE database indicates that this transcript “lncARSR” is a testis-specific transcript, this transcript also is detected in hepatocellular carcinoma (HCC)^{1,2}, ovarian cancer³ and renal cell carcinoma (RCC)⁴. This transcript “lncARSR” composes of four exons with a full length of 591 nt⁴. However, the transcript “lnc-TALC” (ENST00000424980.5) in our work was located on chromosome 9 of human genome and composed of two exons with a full length of 418nt determined by RACE (rapid amplification of cDNA ends) assay (Supplementary Fig. 2C). These used primers and siRNAs in our study accordingly mapped to the sequence of lnc-TALC (Supplementary Table 8 and Table 11).

Supplementary Fig. 2C

(2) This transcript “*LncARSR*” is upregulated in HCC, associated with tumor size and advanced stage, which directly binds to PTEN mRNA, promotes PTEN mRNA degradation and regulates PI3K/Akt pathway¹ and induces dedifferentiation and liver cancer stem cells expansion by targeting STAT3 signaling in HCC cells². *LncARSR* increases SREBP-1c and SREBP-2 expression, involved in the sterol biosynthesis after activation of PI3K/Akt pathway^{5, 6}. *LncARSR* could bind YAP to impede LATS1-induced YAP phosphorylation and to facilitate YAP nuclear translocation⁷, and could be incorporated into exosomes and transmitted to sensitive cells, thus disseminating sunitinib resistance in RCC⁴. In ovarian cancer, *lncARSR* interacts with HuR, upregulates β -catenin expression and then activates Wnt/ β -catenin signaling pathway³. In the present study, *lnc-TALC* that is one of the top 10 upregulated lncRNAs between LN229 and 229R, and knockdown of *lnc-TALC* inhibited the TMZ resistance in TMZ resistant cells. Then, we performed a RACE experiment (5' and 3') for the transcript in this locus, detected the sequence of the transcript *lnc-TALC* and validated its expression in the parental and resistant glioma cells. In addition, we found that *lnc-TALC* trapped miR-20b-3p as a ceRNA, regulated c-Met signaling pathway and increased MGMT expression by remodeling the acetylation of histone H3 in the MGMT promoter regions, indicating that *lnc-TALC* could serve as a therapeutic target to overcome TMZ resistance, enhancing the clinical benefits of TMZ chemotherapy in GBM patients. Above contents have added into the Discussion section in the revised manuscript.

(3) Four different methods including Open reading frame finder from NCBI⁸, phyloCSF⁹,

coding probability from Coding potential assessment tool (CPAT) ¹⁰ and coding potential score from coding potential calculator (CPC) ¹¹ were performed to calculate the coding potential of *lnc-TALC*. Evaluation of putative proteins encoded by *lnc-TALC* using ORF Finder failed to predict a protein of more than 65 amino acids. Nuclear Enriched Abundant Transcript 1 (*NEAT1*) served as a control non-coding gene. Glyceraldehyde-3-phosphate dehydrogenase (*GAPDH*) and β -actin (*ACTB*) served as control coding genes. We defined phyloCSF = 0 ¹², coding probability = 36.4% ¹⁰ and coding potential score = 0 ¹¹ as thresholds and the scores of *lnc-TALC* were all well below the thresholds (phyloCSF scores: -120.6899, coding probability = 2.35%, coding potential score = -0.901102) (Supplementary Fig. 2D-E), indicating the non-coding nature of *lnc-TALC*.

Supplementary Fig. 2D-E

D

E

2. Number of molecules: The relative expression of ceRNA, target and microRNA is an essential point to propose such a mechanism. The authors have not adequately responded to this previous concern as it is not possible to extract molecule numbers per cell from the data provided. The authors need to establish the number of molecules per cell for all three RNAs: *lnc-TALC*, *miR-20b-3p* and *c-MET*. Therefore, they need to perform a dilution series with a plasmid or synthetic polynucleotide for qPCR and then compare the signals from a defined number of cells to estimate the number of molecules per cell.

Response: We feel great thanks for your important suggestions. To confirm *lnc-TALC* as a ceRNA in regulating *c-MET* through competitively binding to *miR-20b-3p* in glioblastoma, we used quantitative real-time PCR to quantify the molecular numbers of *lnc-TALC*, *MET* and *miR-20b-3p* per cell since comparable levels are suggestive of ceRNA function^{13, 14}. After total RNA was isolated from LN229, U251, 551W, HG7, 229R, 251R, 551WR and HG7R cells, 400 ng of RNA was used in the reaction and the fluorescence signals were generated. Molecular number of *miR-20b-3p* per cell was measured in these cells based on a standard curve of synthetic *miR-20b-3p* polynucleotide with known amounts. Molecular numbers of *lnc-TALC* and of *MET* per cell were also measured in these cells based on the standard curves of plasmids containing *lnc-TALC* or *MET* with known amounts (Supplementary Fig. 5G upper). The qRT-PCR analysis combined with standard curves revealed that the average numbers of molecules per cell for *lnc-TALC*, *miR-20b-3p* and *MET* RNAs were at 32, 546, 818 in LN229 cells; 23, 1246, 1231 in U251 cells; 19, 1532, 796 in 551W cells and 65, 945, 1471 in HG7 cells, respectively (Supplementary Fig. 5G middle). The average numbers of molecules per cell for *lnc-TALC*, *miR-20b-3p* and *MET* RNAs were at 324, 289, 4304 in 229R cells; 176, 358, 2118 in 251R cells; 192, 90, 3641 in 551WR cells and 236, 408, 2738 in HG7R cells, respectively (Supplementary Fig. 5G lower). These results are consistent with the previous reports that ceRNA interaction is optimal when the transcript abundance of ceRNAs within a network is near equimolarity^{13, 15, 16, 17, 18}.

Supplementary Fig. 5G

References:

- Li Y, Ye Y, Feng B, Qi Y. Long Noncoding RNA IncARSR Promotes Doxorubicin Resistance in Hepatocellular Carcinoma via Modulating PTEN-PI3K/Akt Pathway. *Journal of cellular biochemistry* 2017, **118**(12): 4498-4507.
- Yang C, Cai WC, Dong ZT, Guo JW, Zhao YJ, Sui CJ, *et al.* IncARSR promotes liver cancer stem cells expansion via STAT3 pathway. *Gene* 2019, **687**: 73-81.
- Shu C, Yan D, Mo Y, Gu J, Shah N, He J. Long noncoding RNA IncARSR promotes epithelial ovarian cancer cell proliferation and invasion by association with HuR and miR-200 family. *American journal of cancer research* 2018, **8**(6): 981-992.
- Qu L, Ding J, Chen C, Wu ZJ, Liu B, Gao Y, *et al.* Exosome-Transmitted IncARSR Promotes Sunitinib Resistance in Renal Cancer by Acting as a Competing Endogenous RNA. *Cancer cell* 2016, **29**(5): 653-668.
- Huang J, Chen S, Cai D, Bian D, Wang F. Long noncoding RNA IncARSR promotes hepatic cholesterol biosynthesis via modulating Akt/SREBP-2/HMGCR pathway. *Life sciences* 2018,

203: 48-53.

6. Zhang M, Chi X, Qu N, Wang C. Long noncoding RNA lncARSR promotes hepatic lipogenesis via Akt/SREBP-1c pathway and contributes to the pathogenesis of nonalcoholic steatohepatitis. *Biochem Biophys Res Commun* 2018, **499**(1): 66-70.
7. Qu L, Wu Z, Li Y, Xu Z, Liu B, Liu F, *et al.* A feed-forward loop between lncARSR and YAP activity promotes expansion of renal tumour-initiating cells. *Nature communications* 2016, **7**: 12692.
8. Nishikawa T, Ota T, Isogai T. Prediction whether a human cDNA sequence contains initiation codon by combining statistical information and similarity with protein sequences. *Bioinformatics (Oxford, England)* 2000, **16**(11): 960-967.
9. Lin MF, Jungreis I, Kellis M. PhyloCSF: a comparative genomics method to distinguish protein coding and non-coding regions. *Bioinformatics (Oxford, England)* 2011, **27**(13): i275-282.
10. Wang L, Park HJ, Dasari S, Wang S, Kocher JP, Li W. CPAT: Coding-Potential Assessment Tool using an alignment-free logistic regression model. *Nucleic acids research* 2013, **41**(6): e74.
11. Kong L, Zhang Y, Ye ZQ, Liu XQ, Zhao SQ, Wei L, *et al.* CPC: assess the protein-coding potential of transcripts using sequence features and support vector machine. *Nucleic acids research* 2007, **35**(Web Server issue): W345-349.
12. Wang Y, He L, Du Y, Zhu P, Huang G, Luo J, *et al.* The long noncoding RNA lncTCF7 promotes self-renewal of human liver cancer stem cells through activation of Wnt signaling. *Cell stem cell* 2015, **16**(4): 413-425.
13. Grelet S, Link LA, Howley B, Obellianne C, Palanisamy V, Gangaraju VK, *et al.* A regulated PNUMS mRNA to lncRNA splice switch mediates EMT and tumour progression. *Nature cell biology* 2017, **19**(9): 1105-1115.
14. Yuan Y, Liu B, Xie P, Zhang MQ, Li Y, Xie Z, *et al.* Model-guided quantitative analysis of microRNA-mediated regulation on competing endogenous RNAs using a synthetic gene circuit. *Proceedings of the National Academy of Sciences of the United States of America* 2015, **112**(10): 3158-3163.
15. Denzler R, Agarwal V, Stefano J, Bartel DP, Stoffel M. Assessing the ceRNA hypothesis with quantitative measurements of miRNA and target abundance. *Molecular cell* 2014, **54**(5): 766-776.
16. Bosson AD, Zamudio JR, Sharp PA. Endogenous miRNA and target concentrations determine susceptibility to potential ceRNA competition. *Molecular cell* 2014, **56**(3): 347-359.
17. Tay Y, Rinn J, Pandolfi PP. The multilayered complexity of ceRNA crosstalk and competition.

Nature 2014, **505**(7483): 344-352.

18. Lin C, Zhang S, Wang Y, Wang Y, Nice E, Guo C, *et al.* Functional Role of a Novel Long Noncoding RNA TTN-AS1 in Esophageal Squamous Cell Carcinoma Progression and Metastasis. *Clinical cancer research : an official journal of the American Association for Cancer Research* 2018, **24**(2): 486-498.

REVIEWERS' COMMENTS:

Reviewer #4 (Remarks to the Author):

The authors have now adequately responded to the previous concerns of the initial reviewer.

Point-by-point response to reviewers' comments

Reviewer #4 (Remarks to the Author):

The authors have now adequately responded to the previous concerns of the initial reviewer.

Response: We feel great thanks for your review work and the referee has no comments to our manuscript.